# The Value Equivalence Principle
# for Model-Based Reinforcement Learning

**Christopher Grimm**
Computer Science & Engineering
University of Michigan
crgrimm@umich.edu

**André Barreto, Satinder Singh, David Silver**
DeepMind
{andrebarreto,baveja,davidsilver}@google.com

## Abstract

Learning models of the environment from data is often viewed as an essential component to building intelligent reinforcement learning (RL) agents. The common practice is to separate the learning of the model from its use, by constructing a model of the environment's dynamics that correctly predicts the observed state transitions. In this paper we argue that the limited representational resources of model-based RL agents are better used to build models that are directly useful for value-based planning. As our main contribution, we introduce the principle of value equivalence: two models are value equivalent with respect to a set of functions and policies if they yield the same Bellman updates. We propose a formulation of the model learning problem based on the value equivalence principle and analyze how the set of feasible solutions is impacted by the choice of policies and functions. Specifically, we show that, as we augment the set of policies and functions considered, the class of value equivalent models shrinks, until eventually collapsing to a single point corresponding to a model that perfectly describes the environment. In many problems, directly modelling state-to-state transitions may be both difficult and unnecessary. By leveraging the value-equivalence principle one may find simpler models without compromising performance, saving computation and memory. We illustrate the benefits of value-equivalent model learning with experiments comparing it against more traditional counterparts like maximum likelihood estimation. More generally, we argue that the principle of value equivalence underlies a number of recent empirical successes in RL, such as Value Iteration Networks, the Predictron, Value Prediction Networks, TreeQN, and MuZero, and provides a first theoretical underpinning of those results.

## 1 Introduction

Reinforcement learning (RL) provides a conceptual framework to tackle a fundamental challenge in artificial intelligence: how to design agents that learn while interacting with the environment [36]. It has been argued that truly general agents should be able to learn a model of the environment that allows for fast re-planning and counterfactual reasoning [32]. Although this is not a particularly contentious statement, the question of *how* to learn such a model is far from being resolved. The common practice in model-based RL is to conceptually separate the learning of the model from its use. In this paper we argue that the limited representational capacity of model-based RL agents is better allocated if the future *use* of the model (*e.g.*, value-based planning) is also taken into account during its construction [22, 15, 13].

Our primary contribution is to formalize and analyze a clear principle that underlies this new approach to model-based RL. Specifically, we show that, when the model is to be used for value-based planning, requirements on the model can be naturally captured by an equivalence relation induced by a set

of policies and functions. This leads to the *principle of value equivalence*: two models are value equivalent with respect to a set of functions and a set of policies if they yield the same updates under corresponding Bellman operators. The policies and functions then become the mechanism through which one incorporates information about the intended use of the model during its construction. We propose a formulation of the model learning problem based on the value equivalence principle and analyze how the set of feasible solutions is impacted by the choice of policies and functions. Specifically, we show that, as we augment the set of policies and functions considered, the class of value equivalent models shrinks, until eventually collapsing to a single point corresponding to a model that perfectly describes the environment.

We also discuss cases in which one can meaningfully restrict the class of policies and functions used to tailor the model. One common case is when the construction of an optimal policy through value-based planning only requires that a model predicts a subset of value functions. We show that in this case the resulting value equivalent models can perform well under much more restrictive conditions than their traditional counterparts. Another common case is when the agent has limited representational capacity. We show that in this scenario it suffices for a model to be value equivalent with respect to appropriately-defined bases of the spaces of representable policies and functions. This allows models to be found with less memory or computation than conventional model-based approaches that aim at predicting all state transitions, such as maximum likelihood estimation. We illustrate the benefits of value-equivalent model learning in experiments that compare it against more conventional counterparts. More generally, we argue that the principle of value equivalence underlies a number of recent empirical successes in RL and provides a first theoretical underpinning of those results [40, 34, 26, 16, 33].

## 2 Background

As usual, we will model the agent's interaction with the environment using a *Markov Decision Process* (MDP) $\mathcal{M} \equiv \langle \mathcal{S}, \mathcal{A}, r, p, \gamma \rangle$ where $\mathcal{S}$ is the state space, $\mathcal{A}$ is the action space, $r(s, a, s')$ is the reward associated with a transition to state $s'$ following the execution of action $a$ in state $s$, $p(s'|s, a)$ is the transition kernel and $\gamma \in [0, 1)$ is a discount factor [30]. For convenience we also define $r(s, a) = \mathbb{E}_{S' \sim p(\cdot|s,a)}[r(s, a, S')]$.

A *policy* is a mapping $\pi : \mathcal{S} \mapsto \mathcal{P}(\mathcal{A})$, where $\mathcal{P}(\mathcal{A})$ is the space of probability distributions over $\mathcal{A}$. We define $\Pi \equiv \{\pi \,|\, \pi : \mathcal{S} \mapsto \mathcal{P}(\mathcal{A})\}$ as the set of all possible policies. The agent's goal is to find a policy $\pi \in \Pi$ that maximizes the *value* of every state, defined as

$$v_\pi(s) \equiv \mathbb{E}_\pi \left[ \sum_{i=0}^{\infty} \gamma^i r(S_{t+i}, A_{t+i}) \,|\, S_t = s \right], \tag{1}$$

where $S_t$ and $A_t$ are random variables indicating the state occupied and the action selected by the agent at time step $t$ and $\mathbb{E}_\pi[\cdot]$ denotes expectation over the trajectories induced by $\pi$.

Many methods are available to carry out the search for a good policy [36, 39]. Typically, a crucial step in these methods is the computation of the value function of candidate policies—a process usually referred to as *policy evaluation*. One way to evaluate a policy $\pi$ is through its *Bellman operator*:

$$\mathcal{T}_\pi[v](s) \equiv \mathbb{E}_{A \sim \pi(\cdot|s), S' \sim p(\cdot|s,A)} \left[ r(s, A) + \gamma v(S') \right], \tag{2}$$

where $v$ is any function in the space $\mathbb{V} \equiv \{f \,|\, f : \mathcal{S} \mapsto \mathbb{R}\}$. It is known that $\lim_{n \to \infty} (\mathcal{T}_\pi)^n v = v_\pi$, that is, starting from any $v \in \mathbb{V}$, the repeated application of $\mathcal{T}_\pi$ will eventually converge to $v_\pi$ [30].

In RL it is generally assumed that the agent does not know $p$ and $r$, and thus cannot directly compute (2). In *model-free* RL this is resolved by replacing $v_\pi$ with an action-value function and estimating the expectation on the right-hand-side of (2) through sampling [35]. In *model-based RL*, the focus of this paper, the agent learns approximations $\tilde{r} \approx r$ and $\tilde{p} \approx p$ and use them to compute (2) with $p$ and $r$ replaced by $\tilde{p}$ and $\tilde{r}$ [36].

## 3 Value equivalence

Given a state space $\mathcal{S}$ and an action space $\mathcal{A}$, we call the tuple $m \equiv (r, p)$ a *model*. Note that a model plus a discount factor $\gamma$ induces a Bellman operator (2) for every policy $\pi \in \Pi$. In this paper we

are interested in computing an approximate model $\tilde{m} = (\tilde{r}, \tilde{p})$ such that the induced operators $\tilde{\mathcal{T}}_\pi$, defined analogously to (2), are good approximations of the true $\mathcal{T}_\pi$. Our main argument is that models should only be distinguished with respect to the policies and functions they will actually be applied to. This leads to the following definition:

**Definition 1** (Value equivalence). *Let $\Pi \subseteq \mathbb{\Pi}$ be a set of policies and let $\mathcal{V} \subseteq \mathbb{V}$ be a set of functions. We say that models $m$ and $\tilde{m}$ are* value equivalent *with respect to $\Pi$ and $\mathcal{V}$ if and only if*

$$\mathcal{T}_\pi v = \tilde{\mathcal{T}}_\pi v \text{ for all } \pi \in \Pi \text{ and all } v \in \mathcal{V},$$

*where $\mathcal{T}_\pi$ and $\tilde{\mathcal{T}}_\pi$ are the Bellman operators induced by $m$ and $\tilde{m}$, respectively.*

Two models are value equivalent with respect to $\Pi$ and $\mathcal{V}$ if the effect of the Bellman operator induced by any policy $\pi \in \Pi$ on any function $v \in \mathcal{V}$ is the same for both models. Thus, if we are only interested in $\Pi$ and $\mathcal{V}$, value-equivalent models are functionally identical. This can be seen as an equivalence relation that partitions the space of models conditioned on $\Pi$ and $\mathcal{V}$:

**Definition 2** (Space of value-equivalent models). *Let $\Pi$ and $\mathcal{V}$ be defined as above and let $\mathcal{M}$ be a space of models. Given a model $m$, the* space of value-equivalent models $\mathcal{M}_m(\Pi, \mathcal{V}) \subseteq \mathcal{M}$ *is the set of all models $\tilde{m} \in \mathcal{M}$ that are value equivalent to $m$ with respect to $\Pi$ and $\mathcal{V}$.*

Let $\mathbb{M}$ be a space of models containing at least one model $m^*$ which perfectly describes the interaction of the agent with the environment. More formally, $m^*$ induces the true Bellman operators $\mathcal{T}_\pi$ defined in (2). Given a space of models $\mathcal{M} \subseteq \mathbb{M}$, often one is interested in models $m \in \mathcal{M}$ that are value equivalent to $m^*$. We will thus simplify the notation by defining $\mathcal{M}(\Pi, \mathcal{V}) \equiv \mathcal{M}_{m^*}(\Pi, \mathcal{V})$.

## 3.1 The topology of the space of value-equivalent models

The space $\mathcal{M}(\Pi, \mathcal{V})$ contains all the models in $\mathcal{M}$ that are value equivalent to the true model $m^*$ with respect to $\Pi$ and $\mathcal{V}$. Since any two models $m, m' \in \mathcal{M}(\Pi, \mathcal{V})$ are equally suitable for value-based planning using $\Pi$ and $\mathcal{V}$, we are free to use other criteria to choose between them. For example, if $m$ is much simpler to represent or learn than $m'$, it can be preferred without compromises.

Clearly, the principle of value equivalence can be useful if leveraged in the appropriate way. In order for that to happen, it is important to understand the space of value-equivalent models $\mathcal{M}(\Pi, \mathcal{V})$. We now provide intuition for this space by analyzing some of its core properties. We refer the reader to Figure 1 for an illustration of the concepts to be discussed in this section. We start with a property that follows directly from Definitions 1 and 2:

**Property 1.** *Given $\mathcal{M}' \subseteq \mathcal{M}$, we have that $\mathcal{M}'(\Pi, \mathcal{V}) \subseteq \mathcal{M}(\Pi, \mathcal{V})$.*

The proofs of all theoretical results are in Appendix A.1. Property 1 states that, given a set of policies $\Pi$ and a set of functions $\mathcal{V}$, reducing the size of the space of models $\mathcal{M}$ also reduces the space of value-equivalent models $\mathcal{M}(\Pi, \mathcal{V})$. One immediate consequence of this property is that, if we consider the space of all policies $\mathbb{\Pi}$ and the space of all functions $\mathbb{V}$, we have one of two possibilities: either we end up with a perfect model or we end up with no model at all. Or, more formally:

**Property 2.** $\mathcal{M}(\mathbb{\Pi}, \mathbb{V})$ *either contains $m^*$ or is the empty set.*

Property 1 describes what happens to $\mathcal{M}(\Pi, \mathcal{V})$ when we vary $\mathcal{M}$ with fixed $\Pi$ and $\mathcal{V}$. It is also interesting to ask what happens when we fix the former and vary the latter. This leads to the next property:

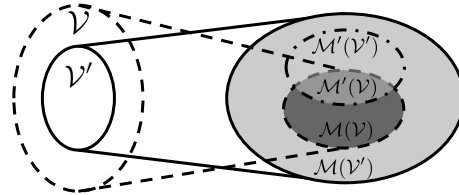

Figure 1: Understanding the space of value-equivalent models for a fixed $\Pi$, $\mathcal{M}' \subset \mathcal{M}$ and $\mathcal{V}' \subset \mathcal{V}$. Denote $\mathcal{M}(\mathcal{V}) \equiv \mathcal{M}(\mathcal{V}, \Pi)$. **Property 1**: $\mathcal{M}'(\mathcal{V}) \subset \mathcal{M}(\mathcal{V})$ and $\mathcal{M}'(\mathcal{V}') \subset \mathcal{M}(\mathcal{V}')$. **Property 3**: $\mathcal{M}(\mathcal{V}) \subset \mathcal{M}(\mathcal{V}')$ and $\mathcal{M}'(\mathcal{V}) \subset \mathcal{M}'(\mathcal{V}')$. **Property 4**: if $m^* \in \mathcal{M}$, then $m^* \in \mathcal{M}(\mathcal{V})$.

**Property 3.** *Given $\Pi' \subseteq \Pi$ and $\mathcal{V}' \subseteq \mathcal{V}$, we have that $\mathcal{M}(\Pi, \mathcal{V}) \subseteq \mathcal{M}(\Pi', \mathcal{V}')$.*

According to Property 3, as we increase the size of $\Pi$ or $\mathcal{V}$ the size of $\mathcal{M}(\Pi, \mathcal{V})$ *decreases*. Although this makes intuitive sense, it is reassuring to know that value equivalence is a sound principle for

model selection, since by adding more policies to $\Pi$ or more values to $\mathcal{V}$ we can progressively restrict the set of feasible solutions. Thus, if $\mathcal{M}$ contains the true model, we eventually pin it down. Indeed, in this case the true model belongs to *all* spaces of value equivalent models, as formalized below:

**Property 4.** *If $m^* \in \mathcal{M}$, then $m^* \in \mathcal{M}(\Pi, \mathcal{V})$ for all $\Pi$ and all $\mathcal{V}$.*

### 3.2 A basis for the space of value-equivalent models

As discussed, it is possible to use the sets $\Pi$ and $\mathcal{V}$ to control the size of $\mathcal{M}(\Pi, \mathcal{V})$. But what exactly is the effect of $\Pi$ and $\mathcal{V}$ on $\mathcal{M}(\Pi, \mathcal{V})$? How much does $\mathcal{M}(\Pi, \mathcal{V})$ decrease in size when we, say, add one function to $\mathcal{V}$? In this section we address this and similar questions.

We start by showing that, whenever a model is value equivalent to $m^*$ with respect to discrete $\Pi$ and $\mathcal{V}$, it is automatically value equivalent to $m^*$ with respect to much larger sets. In order to state this fact more concretely we will need two definitions. Given a discrete set $\mathcal{H}$, we define $\mathrm{span}(\mathcal{H})$ as the set formed by all linear combinations of the elements in $\mathcal{H}$. Similarly, given a discrete set $\mathcal{H}$ in which each element is a function defined over a domain $\mathcal{X}$, we define the *pointwise span* of $\mathcal{H}$ as

$$p\text{-span}(\mathcal{H}) \equiv \left\{ h : h(x) = \sum_i \alpha_{xi} h_i(x) \right\}, \text{ with } \alpha_{xi} \in \mathbb{R} \text{ for all } x \in \mathcal{X}, i \in \{1, \ldots, |\mathcal{H}|\} \quad (3)$$

where $h_i \in \mathcal{H}$. Pointwise span can alternatively be characterized by considering each element in the domain separately: $g \in p\text{-span}(\mathcal{H}) \iff g(x) \in \mathrm{span}\{h(x) : h \in \mathcal{H}\}$ for all $x \in \mathcal{X}$. Equipped with these concepts we present the following result:

**Proposition 1.** *For discrete $\Pi$ and $\mathcal{V}$, we have that $\mathcal{M}(\Pi, \mathcal{V}) = \mathcal{M}(p\text{-span}(\Pi) \cap \mathbb{\Pi}, \mathrm{span}(\mathcal{V}))$.*

Proposition 1 provides one possible answer to the question posed at the beginning of this section: the contraction of $\mathcal{M}(\Pi, \mathcal{V})$ resulting from the addition of one policy to $\Pi$ or one function to $\mathcal{V}$ depends on their effect on $p\text{-span}(\Pi)$ and $\mathrm{span}(\mathcal{V})$. For instance, if a function $v$ can be obtained as a linear combination of the functions in $\mathcal{V}$, adding it to this set will have no effect on the space of equivalent models $\mathcal{M}(\Pi, \mathcal{V})$. More generally, Proposition 1 suggests a strategy to *build* the set $\mathcal{V}$: one should find a set of functions that form a basis for the space of interest. When $\mathcal{S}$ is finite, for example, having $\mathcal{V}$ be a basis for $\mathbb{R}^{|\mathcal{S}|}$ means that the value equivalence principle will apply to every function $\boldsymbol{v} \in \mathbb{R}^{|\mathcal{S}|}$. The same reasoning applies to $\Pi$. In fact, because $p\text{-span}(\Pi)$ grows independently pointwise, it is relatively simple to build a set $\Pi$ that covers the space of policies one is interested in. In particular, when $\mathcal{A}$ is finite, it is easy to define a set $\Pi$ for which $p\text{-span}(\Pi) \supseteq \mathbb{\Pi}$: it suffices to have for every state-action pair $(s, a) \in \mathcal{S} \times \mathcal{A}$ at least one policy $\pi \in \Pi$ such that $\pi(a|s) = 1$. This means that we can apply the value equivalence principle to the entire set $\mathbb{\Pi}$ using $|A|$ policies only.

Combining Proposition 1 and Property 2 we see that by defining $\Pi$ and $\mathcal{V}$ appropriately we can focus on the subset of $\mathcal{M}$ whose models perfectly describe the environment:

**Remark 1.** *If $\mathbb{\Pi} \subseteq p\text{-span}(\Pi)$ and $\mathbb{V} = \mathrm{span}(\mathcal{V})$, then $\mathcal{M}(\Pi, \mathcal{V}) = m^*$ or $\mathcal{M}(\Pi, \mathcal{V}) = \emptyset$.*

We have shown how $\Pi$ and $\mathcal{V}$ have an impact on the number of value equivalent models in $\mathcal{M}(\Pi, \mathcal{V})$; to make the discussion more concrete, we now focus on a specific model space $\mathcal{M}$ and analyze the rate at which this space shrinks as we add more elements to $\Pi$ and $\mathcal{V}$. Before proceeding we define a set of functions $\mathcal{H}$ as *pointwise linearly independent* if $h \notin p\text{-span}(\mathcal{H} \setminus \{h\})$ for all $h \in \mathcal{H}$.

Suppose both $\mathcal{S}$ and $\mathcal{A}$ are finite. In this case a model can be defined as $m = (\boldsymbol{r}, \boldsymbol{P})$, where $\boldsymbol{r} \in \mathbb{R}^{|\mathcal{S}||\mathcal{A}|}$ and $\boldsymbol{P} \in \mathbb{R}^{|\mathcal{S}| \times |\mathcal{S}| \times |\mathcal{A}|}$. A policy can then be thought of as a vector $\boldsymbol{\pi} \in \mathbb{R}^{|\mathcal{S}||\mathcal{A}|}$. We denote the set of all transition matrices induced by transition kernels as $\mathbb{P}$. To simplify the analysis we will consider that $\boldsymbol{r}$ is known and we are interested in finding a model $\tilde{\boldsymbol{P}} \in \mathbb{P}$. In this setting, we write $\mathbb{P}(\Pi, \mathcal{V})$ to denote the set of transition matrices that are value equivalent to the true transition matrix $\boldsymbol{P}^*$. We define the dimension of a set $\mathcal{X}$ as the lowest possible Hamel dimension of a vector-space enclosing some translated version of it: $\dim[\mathcal{X}] = \min_{\mathcal{W}, c \in W(\mathcal{X})} \mathcal{H}\text{-dim}[\mathcal{W}]$ where $W(\mathcal{X}) = \{(\mathcal{W}, c) : \mathcal{X} + c \subseteq \mathcal{W}\}$, $\mathcal{W}$ is a vector-space, $c$ is an offset and $\mathcal{H}\text{-dim}[\cdot]$ denotes the Hamel dimension. Recall that the Hamel dimension of a vector-space is the size of the smallest set of mutually linearly independent vectors that spans the space (this corresponds to the usual notion of dimension, that is, the minimal number of coordinates required to uniquely specify each point). So, under no restrictions imposed by $\Pi$ and $\mathcal{V}$, we have that $\dim[\mathbb{P}] = (|\mathcal{S}| - 1)|\mathcal{S}||\mathcal{A}|$. We now show how fast the size of $\mathbb{P}(\Pi, \mathcal{V})$ decreases as we extend the ranges of $\Pi$ and $\mathcal{V}$:

**Proposition 2.** *Let $\Pi$ be a set of $m$ pointwise linearly independent policies $\boldsymbol{\pi}_i \in \mathbb{R}^{|\mathcal{S}||\mathcal{A}|}$ and let $\mathcal{V}$ be a set of $k$ linearly independent vectors $\boldsymbol{v}_i \in \mathbb{R}^{|\mathcal{S}|}$. Then,*

$$\dim\left[\mathbb{P}(\Pi, \mathcal{V})\right] \leq |\mathcal{S}|\left(|\mathcal{S}||\mathcal{A}| - mk\right).$$

Interestingly, Proposition 2 shows that the elements of $\Pi$ and $\mathcal{V}$ interact in a multiplicative way: when there are $m$ pointwise linearly independent policies, enlarging $\mathcal{V}$ with a single function $v$ that is linearly independent of its counterparts will decrease the bound on the dimension of $\mathbb{P}(\Pi, \mathcal{V})$ by a factor of $m$. This makes intuitive sense if we note that by definition $m \leq |\mathcal{A}|$: for an expressive enough $\Pi$, each $v \in \mathcal{V}$ will provide information about the effect of all actions in $a \in \mathcal{A}$. Conversely, because $\text{span}(\mathcal{V}) = k \leq |\mathcal{S}|$, we can only go so far in pinning down the model when $m < |\mathcal{A}|$—which also makes sense, since in this case we cannot possibly know about the effect of all actions, no matter how big $\mathcal{V}$ is. Note that when $m = |\mathcal{A}|$ and $k = |\mathcal{S}|$ the space $\mathbb{P}(\Pi, \mathcal{V})$ reduces to $\{\boldsymbol{P}^*\}$.

## 4 Model learning based on the value-equivalence principle

We now discuss how the principle of value equivalence can be incorporated into model-based RL. Often in model-based RL one learns a model $\tilde{m} = (\tilde{r}, \tilde{p})$ without taking the space $\mathcal{M}(\Pi, \mathcal{V})$ into account. The usual practice is to cast the approximations $\tilde{r} \approx r$ and $\tilde{p} \approx p$ as optimization problems over a model-space $\mathcal{M}$ that do not involve the sets $\Pi$ and $\mathcal{V}$. Given a space $\mathcal{R}$ of possible approximations $\tilde{r}$, we can formulate the approximation of the rewards as $\text{argmin}_{\tilde{r} \in \mathcal{R}} \ell_r(r, \tilde{r})$, where $\ell_r$ is a loss function that measures the dissimilarity between $r$ and $\tilde{r}$. The approximation of the transition dynamics can be formalized in an analogous way: $\text{argmin}_{\tilde{p} \in \mathcal{P}} \ell_p(p, \tilde{p})$, where $\mathcal{P}$ is the space of possible approximations $\tilde{p}$.

A common choice for $\ell_r$ is

$$\ell_{r,\mathcal{D}}(r, \tilde{r}) \equiv \mathbb{E}_{(S,A) \sim \mathcal{D}}\left[(r(S, A) - \tilde{r}(S, A))^2\right], \tag{4}$$

where $\mathcal{D}$ is a distribution over $\mathcal{S} \times \mathcal{A}$. The loss $\ell_p$ is usually defined based on the principle of *maximum likelihood estimation* (MLE):

$$\ell_{p,\mathcal{D}}(p, \tilde{p}) \equiv \mathbb{E}_{(S,A) \sim \mathcal{D}}\left[\text{D}_{\text{KL}}(p(\cdot|S, A) \,||\, \tilde{p}(\cdot|S, A))\right], \tag{5}$$

where $\text{D}_{\text{KL}}$ is the Kullback-Leibler (KL) divergence. Since we normally do not have access to $r$ and $p$, the losses (4) and (5) are usually minimized using transitions sampled from the environment [38]. There exist several other criteria to approximate $p$ based on state transitions, such as maximum *a posteriori* estimation, maximum entropy estimation, and Bayesian posterior inference [13]. Although we focus on MLE for simplicity, our arguments should extend to these other criteria as well.

Both (4) and (5) have desirable properties that justify their widespread adoption [24]. However, we argue that ignoring the future use of $\tilde{r}$ and $\tilde{p}$ may not always be the best choice [22, 15]. To illustrate this point, we now show that, by doing so, one might end up with an approximate model when an exact one were possible. Let $\mathcal{P}(\Pi, \mathcal{V})$ be the set of value equivalent transition kernels in $\mathcal{P}$. Then,

**Proposition 3.** *The maximum-likelihood estimate of $p^*$ in $\mathcal{P}$ may not belong to a $\mathcal{P}(\Pi, \mathcal{V}) \neq \emptyset$.*

Proposition 3 states that, even when there exist models in $\mathcal{P}$ that are value equivalent to $p^*$ with respect to $\Pi$ and $\mathcal{V}$, the minimizer of (5) may not be in $\mathcal{P}(\Pi, \mathcal{V})$. In other words, even when it is possible to perfectly handle the policies in $\Pi$ and the values in $\mathcal{V}$, the model that achieves the smallest MLE loss will do so only approximately. This is unsurprising since the loss (5) is agnostic of $\Pi$ and $\mathcal{V}$, providing instead a model that represents a compromise across all policies $\mathbb{\Pi}$ and all functions $\mathbb{V}$.

We now define a value-equivalence loss that explicitly takes into account the sets $\Pi$ and $\mathcal{V}$:

$$\ell_{\Pi,\mathcal{V}}(m^*, \tilde{m}) \equiv \sum_{\pi \in \Pi} \sum_{v \in \mathcal{V}} \|\mathcal{T}_\pi v - \tilde{\mathcal{T}}_\pi v\|, \tag{6}$$

where $\tilde{\mathcal{T}}_\pi$ are Bellman operators induced by $\tilde{m}$ and $||\cdot||$ is a norm. Given (6), the problem of learning a model based on the value equivalence principle can be formulated as $\text{argmin}_{\tilde{m} \in \mathcal{M}} \ell_{\Pi,\mathcal{V}}(m^*, \tilde{m})$.

As noted above, we usually do not have access to $\mathcal{T}_\pi$, and thus the loss (6) will normally be minimized based on sample transitions. Let $\mathcal{S}_\pi \equiv \{(s_i^\pi, a_i^\pi, r_i^\pi, \hat{s}_i^\pi)|i = 1, 2, ..., n^\pi\}$ be $n^\pi$ sample transitions associated with policy $\pi \in \Pi$. We assume that the initial states $s_i^\pi$ were sampled according to some

distribution $\mathcal{D}'$ over $\mathcal{S}$ and the actions were sampled according to the policy $\pi$, $a_i^\pi \sim \pi(\cdot | s_i^\pi)$ (note that $\mathcal{D}'$ can be the distribution resulting from a direct interaction of the agent with the environment). When $\| \cdot \|$ appearing in (6) is a $p$-norm , we can write its empirical version as

$$\ell_{\Pi, \mathcal{V}, \mathcal{D}'}(m^*, \tilde{m}) \equiv \sum_{\pi \in \Pi} \sum_{v \in \mathcal{V}} \sum_{s \in \mathcal{S}_\pi'} \left[ \frac{\sum_{i=1}^{n^\pi} \mathbb{1}\{s_i^\pi = s\}(r_i^\pi + \gamma v(\hat{s}_i^\pi))}{\sum_{i=1}^{n^\pi} \mathbb{1}\{s_i^\pi = s\}} - \tilde{\mathcal{T}}_\pi v[s] \right]^p, \qquad (7)$$

where $\mathcal{S}_\pi'$ is a set containing only the initial states $s_i^\pi \in \mathcal{S}_\pi$ and $\mathbb{1}\{\cdot\}$ is the indicator function. We argue that, when we know policies $\Pi$ and functions $\mathcal{V}$ that are sufficient for planning, the appropriate goal for model-learning is to minimize the value-equivalence loss (6). As shown in Proposition 3, the model $\tilde{m}$ that minimizes (4) and (5) may not achieve zero loss on (6) even when such a model exists in $\mathcal{M}$. In general, though, we should not expect there to be a model $\tilde{m} \in \mathcal{M}$ that leads to zero value-equivalence loss. Even then, value equivalence may lead to a better model than conventional counterparts (see Figure 2 for intuition and Appendix A.1.2 for a concrete example).

## 4.1 Restricting the sets of policies and functions

The main argument of this paper is that, rather than learning a model that suits all policies $\Pi$ and all functions $\mathbb{V}$, we should instead focus on the sets of policies $\Pi$ and functions $\mathcal{V}$ that are necessary for planning. But how can we know these sets *a priori*? We now show that it is possible to exploit structure on both the *problem* and the *solution* sides.

First, we consider structure in the *problem*. Suppose we had access to the true model $m^*$. Then, given an initial function $v$, a value-based planning algorithm that makes use of $m^*$ will generate a sequence of functions $\vec{\mathcal{V}}_v \equiv \{v_1, v_2, ...\}$ [10]. Clearly, if we replace $m^*$ with any model in $\mathcal{M}(\Pi, \vec{\mathcal{V}}_v)$, the behavior of the algorithm starting from $v$ remains unaltered. This allows us to state the following:

**Proposition 4.** *Suppose* $v \in \mathcal{V}' \implies \mathcal{T}_\pi v \in \mathcal{V}'$ *for all* $\pi \in \Pi$. *Let* $p$-span$(\Pi) \supseteq \Pi$ *and* span$(\mathcal{V}) = \mathcal{V}'$. *Then, starting from any* $v' \in \mathcal{V}'$, *any* $\tilde{m} \in \mathcal{M}(\Pi, \mathcal{V})$ *yields the same solution as* $m^*$.

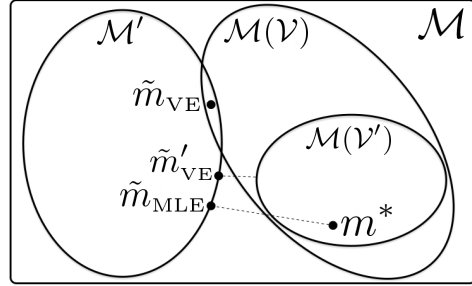

Figure 2: When the hypothesis space $\mathcal{M}'$ and the space of value-equivalent models $\mathcal{M}(\mathcal{V})$ intersect, the resulting model $\tilde{m}_{\mathrm{VE}}$ has zero loss (6), while the corresponding MLE model $\tilde{m}_{\mathrm{MLE}}$ may not (Proposition 3). But even when $\mathcal{M}'$ and $\mathcal{M}(\mathcal{V}')$ do not intersect, the resulting $\tilde{m}_{\mathrm{VE}}'$ can outperform $\tilde{m}_{\mathrm{MLE}}$ with the appropriate choices of $\Pi$ and $\mathcal{V}$, as we illustrate in the experiments of Section 5.

Because $\mathcal{T}_\pi$ are contraction mappings, it is always possible to define a $\mathcal{V}' \subset \mathbb{V}$ such that the condition of Proposition 4 holds: we only need to make $\mathcal{V}'$ sufficiently large to encompass $v$ and the operators' fixed points. But in some cases there exist more structured $\mathcal{V}'$: in Appendix A.1 we give an example of a finite state-space MDP in which a sequence $\boldsymbol{v}_1, \boldsymbol{v}_2 = \mathcal{T}_\pi \boldsymbol{v}_1, \boldsymbol{v}_3 = \mathcal{T}_{\pi'} \boldsymbol{v}_2, ...$ that reaches a specific $k$-dimensional subspace of $\mathbb{R}^{|S|}$ stays there forever. The value equivalence principle provides a mechanism to exploit this type of structure, while conventional model-learning approaches, like MLE, are oblivious to this fact. Although in general we do not have access to $\mathcal{V}'$, in some cases this set will be revealed through the very process of enforcing value equivalence. For example, if $\tilde{m}$ is being learned online based on a sequence $v_1, v_2 = \mathcal{T}_\pi v_1, v_3 = \mathcal{T}_{\pi'} v_2, ...$, as long as the sequence reaches a $v_i \in \mathcal{V}'$ we should expect $\tilde{m}$ to eventually specialize to $\mathcal{V}'$ [13, 33].

Another possibility is to exploit geometric properties of the *value functions* $\vec{\mathcal{V}}_v$. It is known that the set of all value functions of a given MDP forms a polytope $\ddot{\mathcal{V}} \subset \mathbb{V}$ [11]. Even though the sequence $\vec{\mathcal{V}}_v$ an algorithm generates may not be strictly inside the polytope $\ddot{\mathcal{V}}$, this set can still serve as a reference in the definition of $\mathcal{V}$. For example, based on Proposition 1, we may want to define a $\mathcal{V}$ that spans as much of the polytope $\ddot{\mathcal{V}}$ as possible [5]. This suggests that the functions in $\mathcal{V}$ should be actual value functions $v_\pi$ associated with policies $\pi \in \Pi$. In Section 5 we show experiments that explore this idea.

We now consider structure in the *solution*. Most large-scale applications of model-based RL use function approximation. Suppose the agent can only represent policies $\pi \in \tilde{\Pi}$ and value functions $v \in \tilde{\mathcal{V}}$. Then, a value equivalent model $\tilde{m} \in \mathcal{M}(\tilde{\Pi}, \tilde{\mathcal{V}})$ is as good as any model. To build intuition, suppose the agent uses state aggregation to approximate the value function. In this case two models

with the same transition probabilities between clusters of states are indistinguishable from the agent's point of view. It thus makes sense to build $\mathcal{V}$ using piecewise-constant functions that belong to the space of function representable by the agent, $v \in \tilde{\mathcal{V}}$. The following remark generalises this intuition:

**Remark 2.** *Suppose the agent represents the value function using a linear function approximation:* $\tilde{\mathcal{V}} = \{\tilde{v} \,|\, \tilde{v}(s) = \sum_{i=1}^{d} \phi_i(s) w_i\}$, *where* $\phi_i : \mathcal{S} \mapsto \mathbb{R}$ *are fixed features and* $\boldsymbol{w} \in \mathbb{R}^d$ *are learnable parameters. In addition, suppose the agent can only represent policies* $\pi \in \tilde{\Pi}$. *Then, Proposition 1 implies that if we use the features themselves as the functions adopted with value equivalence,* $\mathcal{V} = \{\phi_i\}_{i=1}^{d}$, *we have that* $\mathcal{M}(\tilde{\Pi}, \{\phi_i\}_{i=1}^{d}) = \mathcal{M}(\tilde{\Pi}, \tilde{\mathcal{V}})$. *In other words, models that are value equivalent with respect to the features are indiscernible to the agent.*

According to the remark above, when using linear function approximation, a model that is value equivalent with respect to the approximator's features will perform no worse than any other model. This prescribes a concrete way to leverage the value equivalence principle in practice, since the set of functions $\mathcal{V}$ is automatically defined by the choice of function approximator. Note that, although the remark is specific to linear value function approximation, it applies equally to linear and non-linear models (this is in contrast with previous work showing the equivalence between model-free RL using linear function approximation and model-based RL with a linear model for expected features [27, 38]). The principle of finding a basis for $\tilde{\mathcal{V}}$ also extends to non-linear value function approximation, though in this case it is less clear how to define a set $\mathcal{V}$ that spans $\tilde{\mathcal{V}}$. One strategy is to *sample* the functions to be included in $\mathcal{V}$ from the set $\tilde{\mathcal{V}}$ of (non-linear) functions the agent can represent. Despite its simplicity, this strategy can lead to good performance in practice, as we show next.

## 5 Experiments

We now present experiments illustrating the usefulness of the value equivalence principle in practice. Specifically, we compare models computed based on value equivalence (VE) with models resulting from maximum likelihood estimation (MLE). All our experiments followed the same protocol: $(i)$ we collected sample transitions from the environment using a policy that picks actions uniformly at random, $(ii)$ we used this data to learn an approximation $\tilde{r}$ using (4) as well as approximations $\tilde{p}$ using either MLE (5) or VE (7), $(iii)$ we learned a policy $\tilde{\pi}$ based on $\tilde{m} = (\tilde{r}, \tilde{p})$, and $(iv)$ we evaluated $\tilde{\pi}$ on the actual environment. The specific way each step was carried out varied according to the characteristics of the environment and function approximation used; see App. A.2 for details.

One of the central arguments of this paper is that the value equivalence principle can yield a better allocation of the limited resources of model-based agents. In order to verify this claim, we varied the representational capacity of the agent's models $\tilde{m}$ and assessed how well MLE and VE performed under different constraints. As discussed, VE requires the definition of two sets: $\Pi$ and $\mathcal{V}$. It is usually easy to define a set of policies $\Pi$ such that $p\text{-span}(\Pi) \supseteq \tilde{\Pi}$; since all the environments used in our experiments have a finite action space $\mathcal{A}$, we accomplished that by defining $\Pi = \{\pi^a\}_{a \in \mathcal{A}}$ where $\pi^a(a|s) = 1$ for all $s \in \mathcal{S}$. We will thus restrict our attention to the impact of the set of functions $\mathcal{V}$.

As discussed, one possible strategy to define $\mathcal{V}$ is to use actual value functions in an attempt to span as much as possible of the value polytope $\tilde{\mathcal{V}}$ [5]. Figure 3 shows results of VE when using this strategy. Specifically, we compare VE's performance with MLE's on two well known domains: "four rooms" [37] and "catch" [25]. For each domain, we show two types of results: we either fix the capacity of the model $\tilde{p}$ and vary the size of $\mathcal{V}$ or vice-versa (in the Appendix we show results with all possible combinations of model capacities and sizes of $\mathcal{V}$). Note how the models produced by VE outperform MLE's counterparts across all scenarios, and especially so under stricter restrictions on the model. This corroborates our hypothesis that VE yields models that are tailored to future use.

Another strategy to define $\mathcal{V}$ is to use functions from $\tilde{\mathcal{V}}$, the space of functions representable by the agent, in order to capture as much as possible of this space. In Figure 4 we compare VE using this strategy with MLE. Here we use as domains catch and "cart-pole" [4] (but see Appendix for the same type of result on the four-rooms environment). As before, VE largely outperforms MLE, in some cases with a significant improvement in performance. We call attention to the fact that in cart-pole we used neural networks to represent both the transition models $\tilde{p}$ and the value functions $\tilde{v}$, which indicates that VE can be naturally applied with nonlinear function approximation.

It is important to note the broader significance of our experiments. While our theoretical analysis of value equivalence focused on the case where $\mathcal{M}$ contained a value equivalent model, this is not

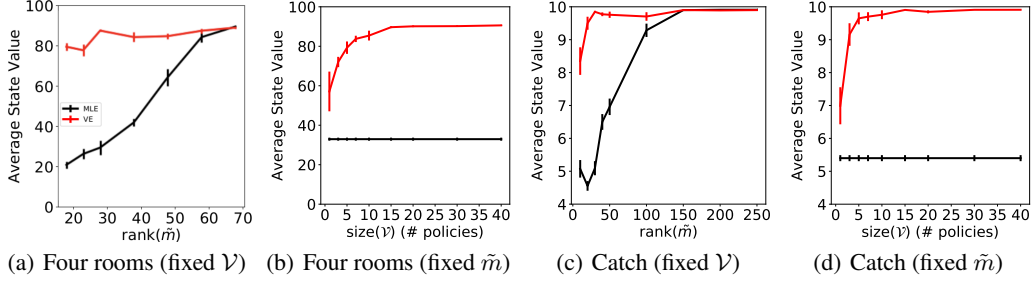

(a) Four rooms (fixed $\mathcal{V}$)  (b) Four rooms (fixed $\tilde{m}$)  (c) Catch (fixed $\mathcal{V}$)  (d) Catch (fixed $\tilde{m}$)

Figure 3: Results with $\mathcal{V}$ composed of true value functions of randomly-generated policies. The models $\tilde{p}$ are rank-constrained transition matrices $\tilde{\boldsymbol{P}} = \boldsymbol{DK}$, with $\boldsymbol{D} \in \mathbb{R}^{|\mathcal{S}| \times k}$, $\boldsymbol{K} \in \mathbb{R}^{k \times |\mathcal{S}|}$, and $k < |\mathcal{S}|$. Error bars are one standard deviation over 30 runs.

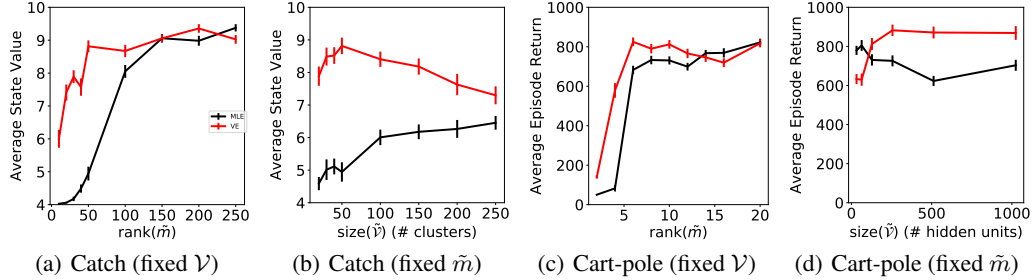

(a) Catch (fixed $\mathcal{V}$)  (b) Catch (fixed $\tilde{m}$)  (c) Cart-pole (fixed $\mathcal{V}$)  (d) Cart-pole (fixed $\tilde{m}$)

Figure 4: Results with $\mathcal{V}$ composed of functions sampled from the agent's representational space $\tilde{\mathcal{V}}$. **(a–b)** Functions in $\mathcal{V}$ are the features of the linear function approximation (state aggregation), as per Remark 2. Models $\tilde{p}$ are rank-constrained transition matrices (*cf.* Figure 3). **(c–d)** Functions in $\mathcal{V}$ are randomly-generated neural networks. Models $\tilde{p}$ are neural networks with rank-constrained linear transformations between layers (Appendix A.2). Error bars are one standard deviation over 30 runs.

guaranteed in practice. Our experiments illustrate that, in spite of lacking such a guarantee, we see a considerable gap in performance between VE and MLE, indicating that VE models still offer a strong benefit. Our goal here was to provide insight into the value equivalence principle; in the next section we point to prior work to demonstrate the utility of value equivalence in large-scale settings.

## 6 The value-equivalence principle in practice

Recently, there have been several successful empirical works that can potentially be understood as applications of the value-equivalence principle, like Silver et al.'s [34] *Predictron*, Oh et al.'s [26] *Value Prediction Networks*, Farquhar et al.'s [16] *TreeQN*, and Schrittwieser et al.'s [33] *MuZero*. Specifically, the model-learning aspect of these prior methods can be understood, with some abuse of notation, as a value equivalence principle of the form $\mathcal{T}v = \tilde{\mathcal{T}}v$, where $\mathcal{T}$ is a Bellman operator applied with the true model $m^*$ and $\tilde{\mathcal{T}}$ is a Bellman operator applied with an approximate model $\tilde{m}$.

There are many possible forms for the operators $\mathcal{T}$ and $\tilde{\mathcal{T}}$. First, value equivalence can be applied to an uncontrolled Markov reward process; the resulting operator $\mathcal{T}_\pi$ is analogous to having a single policy in $\Pi$. Second, it can be applied over $n$ steps, using a Bellman operator $\mathcal{T}_\pi^n$ that rolls the model forward $n$ steps: $\mathcal{T}_\pi^n[v](s) = \mathbb{E}_\pi[R_{t+1} + ... + \gamma^{n-1}R_{t+n} + \gamma^n v_\pi(S_{t+n})|S_t = s]$, or a $\lambda$-weighted average $\mathcal{T}_\pi^\lambda$ [6]. Third, a special case of the $n$-step operator $\mathcal{T}_{a_1...a_n}$ can be applied to an open-loop action sequence $\{a_1, ..., a_n\}$. Fourth, it can be applied to the Bellman optimality operator, $\mathcal{T}_{G_v}$, where $G_v$ is the "greedy" policy induced by $v$ defined as $G_v(a|s) = \mathbb{1}\{a = \operatorname{argmax}_{a'} \mathbb{E}[R + \gamma v(S')|s, a']\}$. This idea can also be extended to an $n$-step greedy search operator, $\mathcal{T}_{G_v^n}[v](s) = \max_{a_1,...,a_n} \mathbb{E}[R_{t+1} + ... + \gamma^{n-1}R_{t+n} + \gamma^n v(S_{t+n})|S_t = s, A_t = a_1, ..., A_{t+n} = a_n]$. Finally, instead of applying value equivalence over a fixed set of value functions $\mathcal{V}$, we can have a set $\mathcal{V}_t$ that varies over time—for example, $\mathcal{V}_t$ can be a singleton with an estimate of the value function of the current greedy policy.

The two operators $\mathcal{T}$ and $\tilde{\mathcal{T}}$ can also differ. For example, on the environment side we can use the optimal value function, which can be interpreted as $\mathcal{T}^\infty v = v^*$ [40, 34], while the approximate operator can be $\tilde{\mathcal{T}}_\pi^\lambda$ [34] or $\tilde{\mathcal{T}}_{G_{v_t}}^n$ [40]. We can also use approximate values $\tilde{\mathcal{T}}v \approx \mathcal{T}v'$ where $v' \approx v$,

for example by applying $n$-step operators to approximate value functions, $\tilde{\mathcal{T}}^n v \approx \mathcal{T}^n v' = \mathcal{T}^n \mathcal{T}^k v = \mathcal{T}^{n+k} v$ [26, 33] or $\tilde{\mathcal{T}}^n v \approx \mathcal{T}^n v' = \mathcal{T}^n \tilde{\mathcal{T}}^k v$ [16], or even to approximate policies, $\tilde{\mathcal{T}}^n v_a \approx \mathcal{T}^n v'_a$ where $v_a = \pi(a|s) \approx \pi'(a|s) = v'_a$ for all $a \in \mathcal{A}$ [33]. The table below characterises the type of value equivalence principle used in prior work. We conjecture that this captures the essential idea underlying each method for model-learning, acknowledging that we ignore many important details.

| Algorithm | Operator $\tilde{\mathcal{T}}$ | Policies $\Pi$ | Functions $\mathcal{V}$ |
|---|---|---|---|
| Predictron [34] | $\tilde{\mathcal{T}}^\lambda v_t$ | None | Value functions for pseudo-rewards |
| VIN [40] | $\tilde{\mathcal{T}}^n_{G_{v_t}} v_t$ | $G_{v_t}$ | Value function |
| TreeQN [16] | $\tilde{\mathcal{T}}^n_{G^n_{v_t}} v_t$ | $G^n_{v_t}$ | Value function |
| VPN [26] | $\tilde{\mathcal{T}}^n_{a_1..a_n} v_t$ | $\{a_1, ..., a_n\} \sim \pi_t$ | Value function |
| MuZero [33] | $\tilde{\mathcal{T}}^n_{a_1...a_n} v_t$ | $\{a_1, ..., a_n\} \sim \pi_t$ | Distributional value bins, policy components |

All of these methods, with the exception of VIN, sample the Bellman operator, rather than computing full expectations (*c.f.* (7)). In addition, all of the above methods jointly learn the state representation alongside a value-equivalent model based upon that representation. Only MuZero includes both many policies and many functions, which may be sufficient to approximately span the policy and function space required to plan in complex environments; this perhaps explains its stronger performance.

# 7 Related work

Farahmand et al.'s [14, 15] *value-aware model learning* (VAML) is based on a premise similar to ours. They study a robust variant of (6) that considers the worst-case choice of $v \in \mathcal{V}$ and provide the gradient when the value-function approximation $\tilde{v}$ is linear and the model $\tilde{p}$ belongs to the exponential family. Later, Farahmand [13] also considered the case where the model is learned iteratively. Both versions of VAML come with finite sample-error upper bound guarantees [14, 15, 13]. More recently, Asadi et al. [2] showed that minimizing the VAML objective is equivalent to minimizing the Wasserstein metric. Abachi et al. [1] applied the VAML principle to policy gradient methods. The theory of VAML is complementary to ours: we characterise the space of value-equivalent models, while VAML focuses on the solution and analysis of the induced optimization problem.

Joseph et al. [22] note that minimizing prediction error is not the same as maximizing the performance of the resulting policy, and propose an algorithm that optimizes the parameters of the model rather than the policy's. Ayoub et al. [3] proposes an algorithm that keeps a set of models that are consistent with the most recent value function estimate. They derive regret bounds for the algorithm which suggest that value-targeted regression estimation is both sufficient and efficient for model-based RL.

More broadly, other notions of equivalence between MDPs have been proposed in the literature [12, 28, 20, 31, 17, 23, 41, 29, 8, 42]. Any notion of equivalence over states can be recast as a form of state aggregation; in this case the functions mapping states to clusters can (and probably should) be used to enforce value equivalence (Remark 2). But the principle of value equivalence is more general: it can be applied with function approximations other than state aggregation and can be used to exploit structure in the problem even when there is no clear notion of state abstraction (Appendix A.1.2).

In this paper we have assumed that the agent has access to a well-defined notion of state $s \in \mathcal{S}$. More generally, the agent only receives observations from the environment and must construct its own state function—that is, a mapping from histories of observations to features representing states. This is an instantiation of the problem known as *representation learning* [43, 21, 9, 18, 45, 44, 19, 7]. An intriguing question which arises in this context is whether a model learned through value equivalence induces a space of "compatible" state representations, which would suggest that the loss (6) could also be used for representation learning. This may be an interesting direction for future investigations.

# 8 Conclusion

We introduced the principle of value equivalence: two models are value equivalent with respect to a set of functions and a set of policies if they yield the same updates of the former on the latter. Value equivalence formalizes the notion that models should be tailored to their future use and provides a mechanism to incorporate such knowledge into the model learning process. It also unifies some important recent work in the literature, shedding light on their empirical success. Besides helping to explain some past initiatives, we believe the concept of value equivalence may also give rise to theoretical and algorithmic innovations that leverage the insights presented.

## Broader impact

The bulk of the research presented in this paper consists of foundational theoretical results about the learning of models for model-based reinforcement learning agents. While applications of these agents can have social impacts depending upon their use, our results merely serve to illuminate desirable properties of models and facilitate the subsequent training of agents using them. In short, this work is largely theoretical and does not present any foreseeable societal impact, except in the general concerns over progress in artificial intelligence.

## Acknowledgements

We would like to thank Gregory Farquhar and Eszter Vertes for the great discussions regarding the value equivalence principle. We also thank the anonymous reviewers for their comments and suggestions to improve the paper.

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
