[Supplementary Material]

# The Value-Equivalence Principle
# for Model-Based Reinforcement Learning
## Supplementary Material

**Christopher Grimm**
Computer Science & Engineering
University of Michigan
crgrimm@umich.edu

**André Barreto, Satinder Singh, David Silver**
DeepMind
{andrebarreto,baveja,davidsilver}@google.com

In this supplement we give details of our theoretical results and experiments that had to be left out of the main paper due to space constraints. We prove our theoretical results and provide a detailed description of our experimental procedure. Importantly, we present an illustrative example showing how value equivalence (VE) may lead to a better solution for a Markov decision process (MDP) than maximum-likelihood estimate (MLE). *We show this to be true both in the exact case, when there exist a value-equivalent model in the model class considered, and in the approximate case, when such a model does not exist in the model class.* Our appendix is organized as follows:

- Section A.1.1 contains derivations of the properties and propositions presented in the main text.
- Section A.1.2 contains a sequence of examples using a toy MDP that illustrate points made in the discussion surrounding Propositions 3 and 4. Moreover, we include an additional result which illustrates a situation in which approximate VE models can outperform the MLE model.
- Section A.2 provides a detailed outline of the pipeline used across our experiments in the main text. We also report several additional results that had to be left out of the main paper due to space constraints.

The numbering of equations, figures and citations resume from what is used in the main paper.

## A Appendix

### A.1 Proofs of theoretical results and illustrative examples

#### A.1.1 Proofs

**Property 1.** *Given $\mathcal{M}' \subseteq \mathcal{M}$, we have that $\mathcal{M}'(\Pi, \mathcal{V}) \subseteq \mathcal{M}(\Pi, \mathcal{V})$.*

*Proof.* This result directly follows from Definitions 1 and 2. $\qquad\square$

**Property 2.** $\mathcal{M}(\mathbb{\Pi}, \mathbb{V})$ *either contains $m^*$ or is the empty set.*

*Proof.* $\mathcal{M}(\mathbb{\Pi}, \mathbb{V}) \subseteq \mathbb{M}(\mathbb{\Pi}, \mathbb{V}) = \{m^*\}$ (Property 1). $\qquad\square$

**Property 3.** *Given $\Pi' \subseteq \Pi$ and $\mathcal{V}' \subseteq \mathcal{V}$, we have that $\mathcal{M}(\Pi, \mathcal{V}) \subseteq \mathcal{M}(\Pi', \mathcal{V}')$.*

*Proof.* We will show the result by contradiction. Suppose there is a model $\tilde{m} \in \mathcal{M}(\Pi, \mathcal{V})$ such that $\tilde{m} \notin \mathcal{M}(\Pi', \mathcal{V}')$. This means that there exists a $\pi \in \Pi'$ and a $v \in \mathcal{V}'$ for which $\tilde{\mathcal{T}}_\pi v \neq \mathcal{T}_\pi v$. But since $\Pi' \subseteq \Pi$ and $\mathcal{V}' \subseteq \mathcal{V}$, it must be the case that $\pi \in \Pi$ and $v \in \mathcal{V}$, which contradicts the claim that $\tilde{m} \in \mathcal{M}(\Pi, \mathcal{V})$. $\qquad\square$

**Property 4.** *If $m^* \in \mathcal{M}$, then $m^* \in \mathcal{M}(\Pi, \mathcal{V})$ for all $\Pi$ and all $\mathcal{V}$.*

*Proof.* $m^* \in \mathcal{M}(\mathbb{\Pi}, \mathbb{V}) \subseteq \mathcal{M}(\Pi, \mathcal{V})$ (Property 3). $\qquad\square$

**Proposition 1.** *For discrete $\Pi$ and $\mathcal{V}$, we have that $\mathcal{M}(\Pi, \mathcal{V}) = \mathcal{M}(p\text{-span}(\Pi) \cap \bar{\Pi}, \text{span}(\mathcal{V}))$.*

*Proof.* Let $\pi \in p\text{-span}(\Pi) \cap \bar{\Pi}$. Based on (3), we know that there exists an $\alpha_s \in \mathbb{R}^{|\Pi|}$ such that $\pi(\cdot|s) = \sum_i \alpha_{si} \pi_i(\cdot|s)$, where $\pi_i \in \Pi$. Thus, for $\tilde{m} \in \mathcal{M}(\Pi, \mathcal{V})$, we can write

$$
\begin{aligned}
\tilde{\mathcal{T}}_\pi[v](s) &= \mathbb{E}_{A \sim \pi(\cdot|s), S' \sim \tilde{p}(\cdot|s,A)} \left[ \tilde{r}(s, A) + \gamma v(S') \right] \\
&= \int \pi(a|s) \mathbb{E}_{S' \sim \tilde{p}(\cdot|s,a)} \left[ \tilde{r}(s, a) + \gamma v(S') \right] da \\
&= \int \sum_i \alpha_{si} \pi_i(a|s) \mathbb{E}_{S' \sim \tilde{p}(\cdot|s,a)} \left[ \tilde{r}(s, a) + \gamma v(S') \right] da \\
&= \sum_i \alpha_{si} \int \pi_i(a|s) \mathbb{E}_{S' \sim \tilde{p}(\cdot|s,a)} \left[ \tilde{r}(s, a) + \gamma v(S') \right] da \\
&= \sum_i \alpha_{si} \mathbb{E}_{A \sim \pi_i(\cdot|s), S' \sim \tilde{p}(\cdot|s,a)} \left[ \tilde{r}(s, a) + \gamma v(S') \right] \\
&= \sum_i \alpha_{si} \mathcal{T}^{\pi_i}[v](s).
\end{aligned}
$$

Let $v \in \text{span}(\mathcal{V})$. We know there is a $\beta \in \mathbb{R}^{|\mathcal{V}|}$ such that $v = \sum_i \beta_i \boldsymbol{v}_i$, with $v_i \in \mathcal{V}$.

$$
\begin{aligned}
\tilde{\mathcal{T}}_\pi[v](s) &= \mathbb{E}_{A \sim \pi(\cdot|s), S' \sim \tilde{p}(\cdot|s,A)} \left[ \tilde{r}(s, A) + \gamma \sum_i \beta_i v_i(S') \right] \\
&= \sum_i \beta_i \mathbb{E}_{A \sim \pi(\cdot|s), S' \sim \tilde{p}(\cdot|s,A)} \left[ \tilde{r}(s, A) + \gamma v_i(S') \right] \\
&= \sum_i \beta_i \tilde{\mathcal{T}}_\pi[v_i](s).
\end{aligned}
$$

$\square$

In order to prove Proposition 2 we will need four lemmas which we state and prove below.

**Lemma 1.** *For arbitrary matrices $\boldsymbol{A} \in \mathbb{R}^{k \times n}, \boldsymbol{C} \in \mathbb{R}^{m \times \ell}$, we can construct a vector-space $\mathcal{B} = \{\boldsymbol{B} \in \mathbb{R}^{n \times m} : \boldsymbol{ABC} = \boldsymbol{0}\}$ where $\boldsymbol{0}$ denotes a $k \times \ell$ matrix of zeros. It follows that*

$$
\mathcal{H}\text{-dim}[\mathcal{B}] = nm - \text{rank}(\boldsymbol{A}) \cdot \text{rank}(\boldsymbol{C}). \tag{8}
$$

*Proof.* We begin by converting the condition $\boldsymbol{ABC} = \boldsymbol{0}$ into a matrix-vector product. Let $\boldsymbol{a}^i$ and $\boldsymbol{c}^j$ denote the i'th row of $\boldsymbol{A}$ and j'th column of $\boldsymbol{C}$ respectively. Observe that $(\boldsymbol{ABC})_{ij} = \boldsymbol{a}^i \boldsymbol{B} \boldsymbol{c}^j = \sum_{x,y} \boldsymbol{a}^i_x \boldsymbol{c}^j_y \boldsymbol{B}_{xy}$, which implies that

$$
\boldsymbol{ABC} = \boldsymbol{0} \iff \sum_{x,y} \boldsymbol{a}^i_x \boldsymbol{c}^j_y \boldsymbol{B}_{xy} = 0 \ \ \forall i \in [k], j \in [\ell] \tag{9}
$$

where $[k]$ denotes $\{1, \ldots, k\}$.

For each $(i, j)$ pair, the above expression is suggestive of a dot-product between two $n \times m$ vectors: a combination of $\boldsymbol{a}^i$ and $\boldsymbol{c}^j$, and a "flattened" version of $\boldsymbol{B}$. Define the former combination of vectors as $\boldsymbol{d}^{ij} = [\boldsymbol{a}^i_1 \boldsymbol{c}^j_1, \boldsymbol{a}^i_1 \boldsymbol{c}^j_2, \cdots, \boldsymbol{a}^i_n \boldsymbol{c}^j_m]^\top \in \mathbb{R}^{nm \times 1}$, and stack them as rows as: $\boldsymbol{D} = [\boldsymbol{d}^{11}, \boldsymbol{d}^{12}, \cdots, \boldsymbol{d}^{nm}]^\top \in \mathbb{R}^{k\ell \times nm}$. To flatten $\boldsymbol{B}$, simply define $\boldsymbol{b} = [\boldsymbol{B}_{11}, \boldsymbol{B}_{12}, \cdots, \boldsymbol{B}_{nm}]^\top \in \mathbb{R}^{nm \times 1}$.

We now have that $\boldsymbol{ABC} = \boldsymbol{0} \iff \boldsymbol{Db} = \boldsymbol{0}$. Moreover, unravelling the matrices in $\mathcal{B}$ does not change the dimension of the space, thus:

$$
\mathcal{H}\text{-dim}[\mathcal{B}] = \mathcal{H}\text{-dim}[\{\boldsymbol{b} \in \mathcal{R}^{nm \times 1} : \boldsymbol{Db} = \boldsymbol{0}\}] = nm - \text{rank}(\boldsymbol{D}) \tag{10}
$$

where the last equality comes from a application of the rank-nullity theorem.

Finally notice that the construction of $\boldsymbol{d}^{ij}$ can be thought of as vertically stacking $n$ copies of $\boldsymbol{c}^j$ each scaled by a different entry in $\boldsymbol{a}^i$. We can also find scaled copies of $\boldsymbol{a}^i$ by $\boldsymbol{c}^j_k$ in $\boldsymbol{d}^{ij}$ by selecting indices from the combined vector at regular intervals of $m$: $\boldsymbol{d}^{ij}_{k+(\ell-1)m} = \boldsymbol{c}^j_k \cdot \boldsymbol{a}^i_\ell$ for $\ell \in \{1, \ldots n\}$.

This means that scaled copies of both $\boldsymbol{a}^i$ and $\boldsymbol{c}^j$ can be found by selecting specific groups of indices in $\boldsymbol{d}^{ij}$. It follows that if $\boldsymbol{a}^1, \ldots, \boldsymbol{a}^n$ are linearly independent then so are $\boldsymbol{d}^{1j}, \ldots, \boldsymbol{d}^{nj}$ for any $j$. And similarly, if $\boldsymbol{c}^1, \ldots, \boldsymbol{c}^m$ are linearly independent then so are $\boldsymbol{d}^{i1}, \ldots, \boldsymbol{d}^{im}$ for any $i$. Hence if $\boldsymbol{a}^1, \ldots \boldsymbol{a}^n$ and $\boldsymbol{c}^1, \ldots, \boldsymbol{c}^m$ are both linearly independent sets, then so is $\boldsymbol{d}^{11}, \boldsymbol{d}^{12}, \ldots, \boldsymbol{d}^{nm}$. Since these $\boldsymbol{a}^i$ and $\boldsymbol{c}^j$ vectors form the rows and columns of rank $n$ and $m$ matrices: $\boldsymbol{A}$ and $\boldsymbol{C}$, their corresponding sets of row and column vectors are linearly independent. Thus we have that $\text{rank}(\boldsymbol{D}) = \text{rank}(\boldsymbol{A}) \cdot \text{rank}(\boldsymbol{C})$, completing the proof. $\square$

**Lemma 2.** *For any $\boldsymbol{c}$ and $\mathcal{Y} + \boldsymbol{c} = \{y + \boldsymbol{c} : y \in \mathcal{Y}\}$ it follows that $\dim[\mathcal{Y} + \boldsymbol{c}] = \dim[\mathcal{Y}]$.*

*Proof.*

$$\dim[\mathcal{Y} + \boldsymbol{c}] = \min_{(\mathcal{V}, \boldsymbol{c}') : \mathcal{Y} + (\boldsymbol{c} + \boldsymbol{c}') \subseteq \mathcal{W}} \mathcal{H}\text{-}\dim[\mathcal{W}] = \min_{(\mathcal{W}, \boldsymbol{c}') : \mathcal{Y} + \boldsymbol{c}' \subseteq \mathcal{W}} \mathcal{H}\text{-}\dim[\mathcal{W}] = \dim[\mathcal{Y}]$$

$\square$

**Lemma 3.** *If $\mathcal{Y}$ is a vector-space then $\mathcal{H}\text{-}\dim[\mathcal{Y}] = \dim[\mathcal{Y}]$.*

*Proof.* Recall the definition of $\dim[\mathcal{Y}]$:

$$\dim[\mathcal{Y}] = \min_{(\mathcal{W}, \boldsymbol{c}) : \mathcal{Y} + \boldsymbol{c} \subseteq \mathcal{W}} \mathcal{H}\text{-}\dim[\mathcal{W}]$$

where $\mathcal{W}$ is a vector-space. By choosing $\mathcal{W} = \mathcal{Y}$ and $\boldsymbol{c} = \boldsymbol{0}$ we see that $\dim[\mathcal{Y}] \leq \mathcal{H}\text{-}\dim[\mathcal{Y}]$.

Suppose then that $\dim[\mathcal{Y}] < \mathcal{H}\text{-}\dim[\mathcal{Y}]$. This implies that there is a vector space $\mathcal{W}$ and offset $\boldsymbol{c}$ with $d = \mathcal{H}\text{-}\dim[\mathcal{W}] < \mathcal{H}\text{-}\dim[\mathcal{Y}]$ and $\mathcal{Y} + \boldsymbol{c} \subseteq \mathcal{W}$. This means that for every $\boldsymbol{y} \in \mathcal{Y}$: $\boldsymbol{y} + \boldsymbol{c} = \sum_{i=1}^{d} \alpha_i^{\boldsymbol{y}} \boldsymbol{w}_i$ for some $\alpha_{1:d}^{\boldsymbol{y}}$ where $\boldsymbol{w}_{1:d}$ are a basis of $\mathcal{W}$. Since $\mathcal{Y}$ is a vector space it must contain the $\boldsymbol{0}$ vector, hence $\boldsymbol{c} = \sum_{i=1}^{d} \alpha_i^{\boldsymbol{0}} \boldsymbol{w}_i$. Accordingly any $\boldsymbol{y} \in \mathcal{Y}$ can be written as $\boldsymbol{y} = \sum_{i=1}^{d} (\alpha_i^{\boldsymbol{y}} - \alpha_i^{\boldsymbol{0}}) \boldsymbol{w}_i$. However, this is a contradiction since $\mathcal{H}\text{-}\dim[\mathcal{W}] < \mathcal{H}\text{-}\dim[\mathcal{Y}]$. Hence $\dim[\mathcal{Y}] = \mathcal{H}\text{-}\dim[\mathcal{Y}]$. $\square$

**Lemma 4.** *If $\mathcal{X} \subseteq \mathcal{Y}$ then $\dim[\mathcal{X}] \leq \dim[\mathcal{Y}]$.*

*Proof.* If $\mathcal{X} \subseteq \mathcal{Y}$ then for any $\boldsymbol{c}$, $\mathcal{X} + \boldsymbol{c} \subseteq \mathcal{Y} + \boldsymbol{c}$. Because of the above, for any vector-space $\mathcal{W}$: $\mathcal{W} \supseteq \mathcal{Y} + \boldsymbol{c} \implies \mathcal{W} \supseteq \mathcal{X} + \boldsymbol{c}$, hence: $\{(\mathcal{W}, c) : \mathcal{X} + \boldsymbol{c} \subset \mathcal{W}\} \supseteq \{(\mathcal{W}, \boldsymbol{c}) : \mathcal{Y} + \boldsymbol{c} \subset \mathcal{W}\}$. Notice that this last set-relation corresponds the set of vector-spaces that $\dim[\cdot]$ is minimizing over for $\mathcal{X}$ and $\mathcal{Y}$ respectively. Hence $\dim[\mathcal{X}] \leq \dim[\mathcal{Y}]$. $\square$

**Proposition 2.** *Let $\Pi$ be a set of $m$ pointwise linearly independent policies $\boldsymbol{\pi}_i \in \mathbb{R}^{|\mathcal{S}||\mathcal{A}|}$ and let $\mathcal{V}$ be a set of $k$ linearly independent vectors $\boldsymbol{v}_i \in \mathbb{R}^{|\mathcal{S}|}$. Then,*

$$\dim[\mathbb{P}(\Pi, \mathcal{V})] \leq |\mathcal{S}| \left( |\mathcal{S}||\mathcal{A}| - mk \right).$$

*Proof.* First note that if $\boldsymbol{\pi}_i \notin p\text{-}\mathrm{span}(\Pi \setminus \{\boldsymbol{\pi}_i\})$ then $\boldsymbol{\pi}_i \notin \mathrm{span}(\Pi \setminus \{\boldsymbol{\pi}_i\})$. Hence, pointwise linear independence implies linear independence.

Since $|\mathcal{S}|$ and $|\mathcal{A}|$ are finite, we can assume that $\mathcal{A} = \{1, \ldots, |\mathcal{A}|\}$ and $\mathcal{S} = \{1, \ldots, |\mathcal{S}|\}$. For any transition probability kernel $\tilde{p}(s'|s, a)$ we can construct matrix $\tilde{\boldsymbol{P}} \in \mathbb{R}^{|\mathcal{S}||\mathcal{A}| \times |\mathcal{S}|}$ with $\tilde{\boldsymbol{P}}_{(a-1)|\mathcal{S}|+s, s'} = \tilde{p}(s'|s, a)$. Denote the constructed matrix corresponding to the true dynamics as $\boldsymbol{P}$. For any $\boldsymbol{\pi}_i$ we can construct a matrix $\boldsymbol{\Pi}_i \in \mathbb{R}^{|S| \times |S||A|}$ with $(\boldsymbol{\Pi}_i)_{s, (a-1)|\mathcal{S}|+s} = \pi_i(a|s)$. Vertically stack these $m$ $\boldsymbol{\Pi}_i$ matrices to construct $\boldsymbol{\Pi} \in \mathbb{R}^{m|S| \times |S||A|}$. Additionally we construct $\boldsymbol{V} \in \mathbb{R}^{|S| \times k}$ with $\boldsymbol{V}_{j, \ell} = (\boldsymbol{v_\ell})_j$. Note that $\mathbb{P}(\Pi, \mathcal{V}) = \{\tilde{\boldsymbol{P}} \in \mathbb{P} : \boldsymbol{\Pi}(\tilde{\boldsymbol{P}} - \boldsymbol{P})\boldsymbol{V} = \boldsymbol{0}\}$. Define the sets $\mathcal{X} = \{\boldsymbol{X} \in \mathbb{R}^{|S||A| \times |S|} : \boldsymbol{P}\boldsymbol{X}\boldsymbol{V} = \boldsymbol{0}\}$ and $\mathcal{Y} = \{\tilde{\boldsymbol{P}} \in \mathbb{R}^{|S||A| \times |S|} : \boldsymbol{\Pi}(\tilde{\boldsymbol{P}} - \boldsymbol{P})\boldsymbol{V} = \boldsymbol{0}\}$.

Note the following three facts:

1. $\dim[\mathcal{X}] = \dim[\mathcal{Y}]$ since our notion of dimension is translation-invariant (Lemma 2).

2. $\dim[\mathcal{X}] = \mathcal{H}\text{-}\dim[\mathcal{X}]$ since $\mathcal{X}$ is a vector-space (Lemma 3).

3. $\mathbb{P}(\Pi, \mathcal{V}) \subseteq \mathcal{Y}$ which implies that $\dim[\mathbb{P}(\Pi, \mathcal{V})] \leq \dim[\mathcal{Y}]$ (Lemma 4).

Taken together this gives us that

$$\dim[\mathbb{P}(\Pi, \mathcal{V})] \leq \dim[\mathcal{Y}] = \mathcal{H}\text{-}\dim[\mathcal{X}].$$

We can now apply Lemma 1 to obtain $\dim[\mathcal{X}] = |S|^2 |A| - k \cdot \mathrm{rank}(\boldsymbol{\Pi})$. Notice that $\mathrm{rank}(\boldsymbol{\Pi}) = \min\{|S||A|, m|S|\}$. Thus $\dim[\mathbb{P}(\Pi, \mathcal{V})] \leq |S|(|S||A| - mk)$ as needed. $\square$

**Proposition 3.** *The maximum-likelihood estimate of $p^*$ in $\mathcal{P}$ may not belong to a $\mathcal{P}(\Pi, \mathcal{V}) \neq \emptyset$.*

*Proof.* Suppose we are trying to estimate a transition matrix $\boldsymbol{P} \in \mathbb{R}^{n \times n}$ and choose to use one parameter $\theta_i \in \mathbb{R}$ per row. Specifically, we parametrize the distribution on the $i$-th row as

$$\tilde{p}_{ii} = \theta_i \text{ and } \tilde{p}_{ij} = (1 - \theta_i)/(n-1), \text{ for } i \neq j, \text{ with } \theta_i \in [0, 1],$$

where $p_{ij} = p(s_j|s_i)$. We can then write the expected likelihood function for $\boldsymbol{\theta} \in \mathbb{R}^n$ as

$$\begin{aligned} m(\boldsymbol{\theta}) &= \sum_i \left[ p_{ii} \ln \theta_i + \sum_{j \neq i} p_{ij} \ln(1 - \theta_i) - \sum_{j \neq i} p_{ij} \ln(n-1) \right] \\ &= \sum_i \left[ p_{ii} \ln \theta_i + (1 - p_{ii}) \ln(1 - \theta_i) - (1 - p_{ii}) \ln(n-1) \right], \end{aligned}$$

which leads to the likelihood equation

$$0 = \frac{\partial m(\boldsymbol{\theta})}{\theta_i} = \frac{p_{ii}}{\theta_i} + \frac{1 - p_{ii}}{\theta_i - 1} = \frac{p_{ii}(\theta_i - 1) + (1 - p_{ii})\theta_i}{\theta_i(\theta_i - 1)} = \frac{\theta_i - p_{ii}}{\theta_i(\theta_i - 1)}.$$

The MLE solution is thus to have $\theta_i = p_{ii}$ for $i = 1, 2, ..., n$. This means that the solution provided by MLE will not be exact if and only if

$$p_{ij} \neq p_{ik} \text{ for any } (i, j, k) \text{ such that } i \neq j \neq k. \tag{11}$$

Now, suppose we have $\mathcal{V} = \{v\}$ with $v_i = 1$ for some $i$ and $v_j = 0$ for $j \neq i$. In this case it is possible to get an exact value-equivalent solution—that is, $\boldsymbol{Pv} = \tilde{\boldsymbol{P}}v$— by making $\theta_i = p_{ii}$ and $\theta_j = 1 - (n-1)p_{ii}$ for $j \neq i$, regardless of whether (11) is true or not. □

**Proposition 4.** *Suppose $v \in \mathcal{V}' \implies \mathcal{T}_\pi v \in \mathcal{V}'$ for all $\pi \in \Pi$. Let $p\text{-span}(\Pi) \supseteq \Pi$ and $\text{span}(\mathcal{V}) = \mathcal{V}'$. Then, starting from any $v' \in \mathcal{V}'$, any $\tilde{m} \in \mathcal{M}(\Pi, \mathcal{V})$ yields the same solution as $m^*$.*

*Proof.* Denote the Bellman operator under a policy that always selects action $a$ as $\mathcal{T}_a$, the greedy Bellman operator as $\mathcal{T}v = \max_a \mathcal{T}_a v$ and the Bellman operator under a policy $\pi$ as $\mathcal{T}_\pi$, as before. Let $\mathcal{T}^{(n)}v$ represent $n$ successive applications of operator $\mathcal{T}$ on value $v$.

Note that for any $v \in \mathbb{V}$ we can construct a $\pi_v(s) = \text{argmax}_a(\mathcal{T}_a v)(s)$ such that $\mathcal{T}v = \max_a \mathcal{T}_a v = \mathcal{T}_{\pi_v}v$. This implies that the greedy Bellman operator is included in the assumption of our proposition:

$$v \in \mathcal{V}' \implies \mathcal{T}v \in \mathcal{V}'. \tag{12}$$

We now begin by showing that:

$$\mathcal{T}^{(n)}v = \tilde{\mathcal{T}}^{(n)}v \in \mathcal{V}' \implies \mathcal{T}^{(n+1)}v = \tilde{\mathcal{T}}^{(n+1)}v \in \mathcal{V}' \tag{13}$$

for any $v \in \mathbb{V}$ and any $n > 0$. Assume that $\mathcal{T}^{(n)}v = \tilde{\mathcal{T}}^{(n)}v \in \mathcal{V}'$. Since $\mathcal{T}^{(n)}v \in \mathcal{V}'$ and $\mathcal{V}' = \text{span}(\mathcal{V})$, we can use use value equivalence to obtain:

$$\mathcal{T}_a\mathcal{T}^{(n)}v = \tilde{\mathcal{T}}_a\mathcal{T}^{(n)}v.$$

for any $a \in \mathcal{A}$. Next, since $\mathcal{T}^{(n)}v = \tilde{\mathcal{T}}^{(n)}v$ we can write:

$$\mathcal{T}_a\mathcal{T}^{(n)}v = \tilde{\mathcal{T}}_a\tilde{\mathcal{T}}^{(n)}v. \tag{14}$$

Since (14) holds for any $a \in \mathcal{A}$, we can write:

$$\mathcal{T}^{(n+1)}v = \max_a \mathcal{T}_a\mathcal{T}^{(n)}v = \max_a \tilde{\mathcal{T}}_a\tilde{\mathcal{T}}^{(n)}v = \tilde{\mathcal{T}}^{(n+1)}v.$$

We know from (12) that the fact that $\mathcal{T}^{(n)}v \in \mathcal{V}'$ implies that $\mathcal{T}^{(n+1)}v \in \mathcal{V}'$. Thus we have shown that (13) is true.

Finally, by choosing $v \in \mathcal{V}'$ and using analogous reasoning as as above, we can show that $\mathcal{T}_a v = \tilde{\mathcal{T}}_a v$ and $\mathcal{T}v = \max_a \mathcal{T}_a v = \max_a \tilde{\mathcal{T}}_a v = \tilde{\mathcal{T}}v$, and since $v \in \mathcal{V}'$, $\tilde{\mathcal{T}}v = \mathcal{T}v \in \mathcal{V}'$. Thus $\tilde{\mathcal{T}}^{(n)}v = \mathcal{T}^{(n)}v$ for all $n \in \mathbb{N}$. This is sufficient to conclude that

$$\tilde{v}^* = \lim_{n \to \infty} \tilde{\mathcal{T}}^{(n)}v' = \lim_{n \to \infty} \mathcal{T}^{(n)}v' = v^*,$$

as needed.

□

### A.1.2   Examples with a simple MDP

Consider the 3 state MDP with states $s_1, s_2, s_3$ and actions $\mathcal{A} = \{a_1, a_2\}$. Transitioning to state $s_1$ always incurs a reward of 1, taking any action in states $s_2$ and $s_3$ always results in transitioning to $s_1$ and taking action $a \in \mathcal{A}$ from $s_1$ transitions among the other states according to action-dependent distribution $(p_{11}^a, p_{12}^a, p_{13}^a)$. This MDP is depicted in Figure 5. We now use this MDP to illustrate several points made in the main text.

Figure 5

**Closure under Bellman updates**  We now address the discussion surrounding Proposition 4 in the main text.

Consider a the following two-dimensional subspace of value functions $\mathcal{R} = \{[x, y, y]^\top : x, y \in \mathbb{R}\}$. We now show that, for the MDP described above, $\mathcal{R}$ exhibits closure under arbitrary Bellman updates.

For an arbitrary policy $\pi : \mathcal{S} \mapsto \mathcal{P}(\mathcal{A})$ the Bellman update for a value function $\boldsymbol{v} \in \mathbb{R}^3$ is given by $\mathcal{T}^\pi \boldsymbol{v} = \boldsymbol{R}^\pi + \gamma \boldsymbol{P}^\pi \boldsymbol{v}$ where

$$\boldsymbol{R}^\pi = \begin{bmatrix} \sum_{a \in \mathcal{A}} \pi(a|s_1) p_{11}^a \\ 1 \\ 1 \end{bmatrix}, \ \boldsymbol{P}^\pi = \begin{bmatrix} \sum_{a \in \mathcal{A}} \pi(a|s_1) p_{11}^a & \sum_{a \in \mathcal{A}} \pi(a|s_2) p_{12}^a & \sum_{a \in \mathcal{A}} \pi(a|s_3) p_{13}^a \\ 1 & 0 & 0 \\ 1 & 0 & 0 \end{bmatrix}$$

Suppose $\boldsymbol{v} \in \mathcal{R}$, then $\boldsymbol{v} = [a, b, b]^\top$ for some $a, b \in \mathbb{R}$. Notice that for such a value function the following holds:

$$\mathcal{T}^\pi v = \begin{bmatrix} \boldsymbol{R}_1^\pi + \gamma[a \boldsymbol{P}_{11}^\pi + b(1 - \boldsymbol{P}_{11}^\pi)] \\ 1 + \gamma a \\ 1 + \gamma a \end{bmatrix} \in \mathcal{R},$$

thus we have illustrated that the two-dimensional subspace $\mathcal{R}$ is closed under arbitrary Bellman updates in our 3 state MDP. This means that, once a sequence $\boldsymbol{v}_1, \boldsymbol{v}_2 = \mathcal{T}_\pi \boldsymbol{v}_1, \boldsymbol{v}_3 = \mathcal{T}_{\pi'} \boldsymbol{v}_2...$ reaches a $\boldsymbol{v}_i \in \mathcal{R}$, it stays in $\mathcal{R}$. We can then exploit this property finding value-equivalent models with respect to $\mathcal{R}$, as we show next.

**A model class for which *exact* VE outperforms MLE**  We now provide an example of the scenario discussed around Proposition 3 in the main text by examining the setting where a model, from a restricted class, must be learned to approximate the dynamics of our MDP. We restrict our model class by requiring that for each action $a \in \mathcal{A}$ we represent $(p_{11}^a, p_{12}^a, p_{13}^a)$ as $((1 - \theta^a)/2, \theta^a, (1 - \theta^a)/2)$. Before continuing we note a few properties of value functions of our MDP. Notice that for any $\boldsymbol{v}^\pi$ we can write:

$$\boldsymbol{v}_1^\pi = \sum_{a \in \mathcal{A}} \pi(a|s_1)[p_{11}^a(1 + \gamma \boldsymbol{v}_1^\pi) + (1 - p_{11}^a)(\gamma^2 \boldsymbol{v}_1^\pi)],$$
$$\boldsymbol{v}_2^\pi = 1 + \gamma \boldsymbol{v}_1^\pi,$$
$$\boldsymbol{v}_3^\pi = 1 + \gamma \boldsymbol{v}_1^\pi,$$

which illustrates that $v^\pi$ *exclusively depends* on the value of $\boldsymbol{P}_{11}^\pi \equiv \sum_{a \in \mathcal{A}} \pi(a|s_1) p_{11}^a$.

First we consider the MLE solution to this problem: it can be easily shown (see the proof of Proposition 3) that, for the model class defined above, $\theta^a = p_{12}^a$ for all $a \in \mathcal{A}$ maximizes the likelihood. However notice that this implies that our approximation of $p_{11}^a$ equals $(1 - p_{12}^a)/2$ which is clearly not true in general. Thus, there are settings of $(p_{11}^a, p_{12}^a, p_{13}^a)$ and policies for which the value function produced by MLE, $\tilde{\boldsymbol{v}}^\pi$, is not equivalent to the true value function $\boldsymbol{v}^\pi$.

Next we consider learning a value-equivalent model with the same restricted model class. Suppose we wish our model to be value equivalent to value $\boldsymbol{v} = [1, 0, 0]^\top$ and all policies.

Note that any VE model with respect to $\mathcal{V} = \{\boldsymbol{v}\}$: $\{\tilde{\boldsymbol{P}}^a\}_{a \in \mathcal{A}}$, must satisfy $\tilde{\boldsymbol{P}}^a v = \boldsymbol{P}^a v$. By requiring value equivalence with just $v$ we have:

$$\tilde{\boldsymbol{P}}^a \boldsymbol{v} = \begin{bmatrix} \tilde{p}_{11}^a \\ \tilde{p}_{21}^a \\ \tilde{p}_{31}^a \end{bmatrix} = \begin{bmatrix} p_{11}^a \\ 1 \\ 1 \end{bmatrix} = \boldsymbol{P}^a \boldsymbol{v}$$

which implies that $\tilde{p}_{11}^a = p_{11}, \tilde{p}_{21}^a = \tilde{p}_{31}^a = 1$ and $\tilde{p}_{22}^a = \tilde{p}_{23}^a = \tilde{p}_{32}^a = \tilde{p}_{33}^a = 0$ for all $a \in \mathcal{A}$.

Taking these constraints together restricts the class of VE models to those of the form:

$$\tilde{\boldsymbol{P}} = \begin{bmatrix} p_{11}^a & \tilde{p}_{12}^a & \tilde{p}_{13}^a \\ 1 & 0 & 0 \\ 1 & 0 & 0 \end{bmatrix}$$

where $\tilde{p}_{1i}^a$ are "free variables" for all $i = 2, 3$ and $a \in \mathcal{A}$.

Notice that when $p_{11}^a \leq 0.5$ for all $a \in \mathcal{A}$, we can find a value equivalent model by setting: $(1 - \theta^a)/2 = p_{11}^a$. This means that the values produced by these value equivalent models exactly match those of the environment: $\tilde{\boldsymbol{v}}^\pi = \boldsymbol{v}^\pi$ for all $\pi$ (and thus the solution of this model also coincides with the optimal value function, $\tilde{\boldsymbol{v}}^* = \boldsymbol{v}^*$).

**A model class for which *approximate* VE outperforms MLE**    In the previous example we showed that it is possible to have an MDP and a restricted model class such that VE models are able to perfectly estimate $\boldsymbol{v}^*$ while MLE models fail to do so. Notice that in this example a value equivalent model *actually existed*, which is not guaranteed in general. We now show a related example where, in spite of an exactly value equivalent model not existing, an agent trained using an *approximate* value equivalent model will outperform its MLE counterpart.

We use our example MDP from before, shown in Figure 5, and denote its actions $\mathcal{A} = \{a, b\}$ for later notational convenience. We set our environment's transition dynamics accordingly: $p^a \equiv (p_{11}^a, p_{12}^a, p_{13}^a) = (0.6, 0.4, 0.0)$ and $p^b \equiv (p_{11}^b, p_{12}^b, p_{13}^b) = (0.4, 0.2, 0.4)$. We also use the same model class as above: $(\tilde{p}_{11}^i, \tilde{p}_{12}^i, \tilde{p}_{13}^i) = (0.5(1 - \theta^i), \theta^i, 0.5(1 - \theta^i))$ for each $i \in \mathcal{A}$, being mindful of the boundary conditions $\theta^i \in [0, 1]$.

As we saw in the previous example, the MLE estimator for this problem will produce the following approximations: $p_{\text{MLE}}^a = (0.3, 0.4, 0.3)$, $p_{\text{MLE}}^b = (0.4, 0.2, 0.4)$.

We now consider what an approximate VE model will produce using the same value as before: $v = [1, 0, 0]^\top$ and all policies. Recall that we're optimizing the following loss:

$$\begin{aligned} \sum_{j \in \{a,b\}} \sum_{i=1}^{3} ((\tilde{\boldsymbol{P}}^j v)_i - (\boldsymbol{P}^j v)_i))^2 &= \sum_{j \in \{a,b\}} (\tilde{p}_{11}^j - p_{11}^j)^2 + ((\tilde{p}_{12}^j + \tilde{p}_{13}^j) - (p_{12}^j + p_{13}^j))^2 \\ &= \sum_{j \in \{a,b\}} (\tilde{p}_{11}^j - p_{11}^j)^2 + ((1 - \tilde{p}_{11}^j) - (1 - p_{11}^j))^2 \\ &= \sum_{j \in \{a,b\}} 2(\tilde{p}_{11}^j - p_{11}^j)^2 \\ &= 2(\tilde{p}_{11}^a - p_{11}^a)^2 + 2(\tilde{p}_{11}^b - p_{11}^b)^2. \end{aligned}$$

The form of this loss indicates that VE will attempt to minimize the MSE of $\tilde{p}_{11}^a$ and $\tilde{p}_{11}^b$ separately. Notice that for action $a$, we cannot perfectly estimate $p_{11}$ due to the boundary conditions on $\theta^a$. However, VE will still find the closest possible $\tilde{p}_{11}$ that respects the boundary condition, giving: $\tilde{p}_{\text{VE}}^a = (0.5, 0.0, 0.5)$, $\tilde{p}_{\text{VE}}^b = (0.4, 0.2, 0.4)$.

We now display these models together in the following table:

|      | $\tilde{p}_{11}^a$ | $\tilde{p}_{12}^a$ | $\tilde{p}_{13}^a$ | $\tilde{p}_{11}^b$ | $\tilde{p}_{12}^b$ | $\tilde{p}_{13}^b$ |
|------|------|------|------|------|------|------|
| MDP  | 0.6  | 0.4  | 0.0  | 0.4  | 0.2  | 0.4  |
| MLE  | 0.3  | 0.4  | 0.3  | 0.4  | 0.2  | 0.4  |
| VE   | 0.5  | 0.0  | 0.5  | 0.4  | 0.2  | 0.4  |

Notice that when optimally planning on this MDP, an agent can obtain the most reward by transitioning from $s_1$ to $s_1$ as often as possible. The agent can do this taking the action among $\{a, b\}$ that is mostly likely to induce a self-transition each time it is at $s_1$. In the true environment and the VE model this action is $a$. However, notice that the MLE model would instead prefer the sub-optimal action $b$, since $(\tilde{p}_{\text{MLE}}^b)_{11} > (\tilde{p}_{\text{MLE}}^a)_{11}$.

This is a concrete example where VE outperforms MLE even though there is no value-equivalent models in the model class considered (that is, VE can be enforced only approximately).

## A.2 Experimental details

(a) Catch        (b) Four Rooms        (c) Cart-pole

Figure 6: **(a) Catch:** the agent has three actions corresponding to moving a paddle (orange) left, right and staying in place. Upon initialization, a ball (blue) is placed at a random square at the top of the environment and at each step it descends by one unit. Upon reaching the bottom of the environment the ball is returned to a random square at the top. The agent receives a reward of $1.0$ if it moves its paddle and intercepts the ball. **(b) Four Rooms:** the agent (orange) has four actions corresponding to up, down, left and right movement. When the agent takes an action, it moves in its intended direction with 90% of the time and in an random other direction otherwise. There is a rewarding square in the right top corner (green). If the agent transitions into this square it receives a reward of $1.0$. **(c) Cart-pole:** In Cart-pole, the agent may choose between three actions: pushing the cart to the left, right or not pushing the cart. There is a pole balanced on top of the cart that is at risk of tipping over. The agent is incentivized to keep the pole up-right through a reward of $\cos(\theta)$ at each step where $\theta$ is the angle of the pole ($\theta = 0$ implies the pole is perfectly up-right). If the pole's height drops below a threshold, the episode terminates and the agent receives a reward of $0.0$. The cart itself is resting on a table; if it falls off the table, the episode similarly terminates with a reward of $0.0$.

### A.2.1 Environment description

The environments used in our experiments are described in depth in Figure 6. In both Catch and Four Rooms a tabular representation is employed in which each of the environment's finitely many states (250 and 68, respectively) is represented by an index. In Cart-pole we have a continuous state space $\mathcal{S} \subset \mathbb{R}^5$ (so $|\mathcal{S}| = \infty$). Each state $s \in \mathbb{R}^5$ consists of the cart position, cart velocity, sine / cosine of pole angle, and pole's angular velocity.

### A.2.2 Experimental pipeline

As mentioned in the main text, a common experimental pipeline is used across all of our results, with slight variations depending upon the experiment type and environment. This pipeline is described at a high-level below:

  (i) **Data collection:** Data is collected using a policy which selects actions uniformly at random.

 (ii) **Model training:** The collected data is used to train a model.

(iii) **Policy construction:** The model is used to produce a policy.

(iv) **Policy evaluation:** The policy is evaluated to assess the quality of the model.

We now discuss steps (ii), (iii) and (iv) in detail.

**(ii) Model training** All of our experiments involve restricting the capacity of the class of models that the agent can represent: $\mathcal{M}$. In general we restrict the rank of the models in $\mathcal{M}$, but, depending upon the nature of the model, this restriction is carried in different ways.

    1. **Tabular models:** On domains with $|S| < \infty$, we employ tabular models. In what follows, $n \times m$ matrices referred to as "row-stochastic" are ensured to be as such by the following parameterization:

      (a) A matrix $\boldsymbol{F} \in \mathbb{R}^{n \times m}$ is sampled with entries $\boldsymbol{F}_{ij} \sim \text{Uniform}([-1, 1])$.

(b) A new matrix $\boldsymbol{P}_F$ is produced by applying row-wise softmax operations with temperature $\tau = 1$ to $\boldsymbol{F}$:

$$(\boldsymbol{P}_F)_{ij} = \frac{\exp(\boldsymbol{F}_{ij})}{\sum_k \exp(\boldsymbol{F}_{ik})}.$$

Here, $\boldsymbol{F}$ can be thought of as the parameters of $\boldsymbol{P}_F$, which often will suppress as $\tilde{\boldsymbol{P}}$ for clarity.

That is, a model is represented by $|A|$ $|S| \times |S|$ row-stochastic matrices: $\tilde{\boldsymbol{P}}^1, \ldots, \tilde{\boldsymbol{P}}^{|A|}$. We ensure that each of these matrices has rank $k$ by factoring it as follows: $\tilde{\boldsymbol{P}}^a = \boldsymbol{D}^a \boldsymbol{K}^a$ where $\boldsymbol{D}^a \in \mathbb{R}^{|S| \times k}$, $\boldsymbol{K}^a \in \mathbb{R}^{k \times |S|}$ and both are row-stochastic as well.

2. **Neural network models:** On domains with $|S| = \infty$ we instead use a neural network parameterized by $\theta$: $f_\theta : (\mathcal{S}, \mathcal{A}) \mapsto (\mathcal{S}, \mathbb{R})$. $f_\theta$ takes a state and action as input and outputs an approximation of the expected next state and next reward. As an analogue to the rank restriction applied in the tabular case, we restrict the rank of weight matrices in all fully-connected layers in $f_\theta$. Denote a fully-connected layer in $f_\theta$ as $L(x) = \sigma(Wx + b)$ where $\sigma(\cdot)$ is an activation function, $W$ is a weight matrix and $b$ is a bias term. We restrict $f_\theta$ by replacing each $L(x)$ with $L_k(x) = \sigma((DK)x + b)$ where $D, K \in \mathbb{R}^{|S| \times k}, \mathbb{R}^{k \times |S|}$.

The models with the restrictions above are trained based on data collected by a policy that selects actions uniformly at random. With a small abuse of notation, denote the collected data as $\mathcal{D} = (s_i, a_i, r_i, s_i')_{i=1}^N$. We will now describe how this data is used to train models in different contexts.

1. **Tabular models:** When training a tabular model with capacity restricted to rank $k$, we use the following expressions:

   (a) **Reward**: In our experiments rewards are represented in the same way for both VE and MLE models:

   $$\tilde{R}_{s,a} = \frac{\sum_{i=1}^N r_i \mathbb{1}\{s_i = s, a_i = a\}}{\sum_{i=1}^N \mathbb{1}\{s_i = s, a_i = a\}},$$

   where $\mathbb{1}\{\cdot\}$ is the indicator function.

   (b) **Transition dynamics (MLE)**: To learn the transition dynamics we first parameterize $\tilde{\boldsymbol{P}}^a = \boldsymbol{D}^a \boldsymbol{K}^a$ for all $a \in \mathcal{A}$, where $\boldsymbol{D}^a$ and $\boldsymbol{K}^a$ are row-stochastic matrices (see item 1 in the section "Restricting Model Capacity" above). Because we are assuming $\mathcal{S}$ to be finite, we can identify each state $s \in \mathcal{S}$ by an index. Let $\delta(s) \in \{1, ..., |S|\}$ be an index that uniquely identifies state $s$. We then compute $\tilde{\boldsymbol{P}}^a = \boldsymbol{D}^a \boldsymbol{K}^a$ by minimizing the following loss with respect to $\boldsymbol{D}^a$ and $\boldsymbol{K}^a$:

   $$\tilde{\ell}_{p,\mathcal{D}}(\boldsymbol{P}^a, \tilde{\boldsymbol{P}}^a) \equiv -\sum_{i=1}^N \mathbb{1}\{a_i = a\} \log\left[(\boldsymbol{D}^a \boldsymbol{K}^a)_{\delta(s_i)\delta(s_i')}\right],$$

   where $(\boldsymbol{D}^a \boldsymbol{K}^a)_{ij}$ is the element in the $i$-th row and $j$-th column of matrix $\boldsymbol{D}^a \boldsymbol{K}^a$. Note that the expression above is the empirical version of expression (5) in the paper [15].

   (c) **Transition dynamics (VE)**: In the VE setting we have a set of value functions and policies: $\mathcal{V}$ and $\Pi$. We have one transition matrix $\tilde{\boldsymbol{P}}^\pi$ associated with each policy $\pi \in \Pi$. As discussed in Section 5, in our experiments we used $\Pi = \{\pi^a\}_{a \in \mathcal{A}}$, where $\pi^a(a|s) = 1$ for all $s \in \mathcal{S}$. Thus, we end up with the same parameterized probability matrices as above: $\tilde{\boldsymbol{P}}^a = \boldsymbol{D}^a \boldsymbol{K}^a$. Let $\mathcal{D}_{ia} \subseteq \mathcal{D}$ be the sample transitions starting in state $i$ where action $a$ was taken, that is, $(s_j, a_j, r_j, s_j') \in \mathcal{D}_{ia}$ if and only if $\delta(s_j) = i$ and $a_j = a$. We computed $\tilde{\boldsymbol{P}}^a = \boldsymbol{D}^a \boldsymbol{K}^a$ by minimizing the following loss with respect to $\boldsymbol{D}^a$ and $\boldsymbol{K}^a$:

   $$\ell_{\pi^a, \mathcal{V}, \mathcal{D}}(\boldsymbol{P}^a, \tilde{\boldsymbol{P}}^a) \equiv \sum_{i,a} \sum_{\boldsymbol{v} \in \mathcal{V}} \left( \frac{1}{|\mathcal{D}_{ia}|} \sum_{(s,a,r,s') \in \mathcal{D}_{ia}} v_{\delta(s')} - \sum_j (\boldsymbol{D}^a \boldsymbol{K}^a)_{ij} v_j \right)^2.$$

   Note that the expression above corresponds to equation (7) when learning transition matrices associated with policies $\{\pi^a\}_{a \in \mathcal{A}}$ in an environment with finite state space $\mathcal{S}$ (where states $s$ can be associated with an index $i$) and $p = 2$.

2. **Neural network models**: When training a neural network model with capacity restrictions construct a network $f_\theta : (\mathcal{S}, \mathcal{A}) \mapsto (\mathcal{S}, \mathbb{R})$. The network is fully connected and takes the concatenation of $\mathcal{S}$ with the one-hot representation of $\mathcal{A}$ as input. For a given $(s, a)$ pair we denote it's output as $\tilde{s}'_{s,a}, \tilde{r}'_{s,a} = f_\theta(s, a)$. In all cases we train the neural network model by sampling mini-batches uniformly from $\mathcal{D}$. It is important to note that we only use these neural network models on deterministic domains (e.g., Cart-pole) meaning that the output of the model, $\tilde{s}'$ represents a single state rather than an expectation over states.

   (a) **Reward:** For both VE and MLE models we train our neural network models to accurately predict the reward associated with each state action transition:

   $$\ell_{r,\mathcal{D}}(\theta) = \sum_{i=1}^{N} (\tilde{r}_{s_i,a_i} - r_i)^2.$$

   (b) **Transition dynamics (MSE):** We learn models by encouraging $f_\theta$ to accurately predict the next state:

   $$\ell_{s',\mathcal{D}}(\theta) = \sum_{i=1}^{N} (\tilde{s}'_{s_i,a_i} - s'_i)^2.$$

   (c) **Transition dynamics (VE):** For VE models use (7), disregarding reward terms to give:

   $$\ell_{\mathcal{V},\mathcal{D}}(\theta) = \sum_{i=1}^{n} \sum_{v \in \mathcal{V}} (v(\tilde{s}_{s_i,a_i}) - v(s'_i))^2.$$

**(iii) Policy construction**   In each experiment we present, after a model is constructed, we subsequently use it to construct a policy. The manner in which we do this varies based upon the type of the experiment and the nature of the environment. There are three mechanisms for constructing policies from models:

1. **Value iteration:** For experiments with $\mathcal{V} = \mathbb{V}$ (which are performed only with tabular models), we use the learned model $\tilde{m} = (\tilde{r}, \tilde{p})$ to perform value iteration until convergence, yielding $\tilde{v}^*$ [30]. Here $\tilde{v}^*$ represents the optimal value function of the model $\tilde{m}$. We then produce a policy according to $\pi(s) = \text{argmax}_a (\tilde{r}(s, a) + \gamma \sum_{s'} \tilde{p}(s'|s, a)\tilde{v}^*(s'))$.

2. **Approximate policy iteration with least squares temporal-difference learning (LSTD):** For experiments on environments with finite $\mathcal{S}$ and $\mathcal{V} = \tilde{\mathcal{V}}$ we used policy iteration combined with least square policy evaluation using basis $\{\phi_i\}_{i=1}^{d}$. Specifically, each iteration of policy iteration involved the following steps:

   (a) Collect experience tuples using the previous policy, $\pi$, leading to $\mathcal{D} = (s_i, a_i, r_i, s'_i)_{i=1}^{n}$.
   (b) Replace the reward and next-states with those predicted by the model: $\tilde{r}_i, \tilde{s}'_i = f_\theta(s_i, a_i)$, leading to $\mathcal{D}' = (s_i, a_i, \tilde{r}_i, \tilde{s}'_i)_{i=1}^{n}$.
   (c) Learn $v_w(s) = \sum_{i=1}^{d} w_i \phi_i(s) \approx v_\pi$ using LSTD with $\mathcal{D}'$.
   (d) Construct a new policy $\pi(s) = \text{argmax}_a (\tilde{r}_{s,a} + \gamma v_w(\tilde{s}'_{s,a}))$ where $\tilde{r}_{s,a}, \tilde{s}'_{s,a}$ are sampled from the trained model conditioned on state $s$ and action $a$.

   This procedure is repeated for a fixed number of iterations.

3. **Deep Q-networks (DQN):** For experiments with $\mathcal{V} = \tilde{\mathcal{V}}$ and infinite $\mathcal{S}$ we use Double Q-Learning to produce policies. We incorporate our learned model, $f_\theta$, by replacing elements in the replay buffer $(s, a, r, s')$ with $(s, a, \tilde{r}_{s,a}, \tilde{s}'_{s,a})$ where $\tilde{r}_{s,a}, \tilde{s}'_{s,a} = f_\theta(s, a)$.

**(iv) Policy evaluation**   There are two methods to evaluate the policies resulting from the policy construction stage described above:

1. For policies produced using value iteration or policy iteration plus LSTD the ensuing policy, $\pi$, is exactly evaluated on the true environment, yielding $v_\pi(s)$. Then the average value of $v_\pi(s)$ over all states is reported.

2. For policies produced using DQN, the average return over the last 100 episodes of training is reported.

### A.2.3 Classes of experiments

In addition to varying the capacity of $\mathcal{M}$, there are two primary classes of experiments that were run in our paper that assess different choices of $\mathcal{V}$. We distinguish between these two classes below:

**span$(\mathcal{V}) \approx \ddot{\mathcal{V}}, \tilde{\mathcal{V}} = \mathbb{V}, \mathbf{\Pi} = \mathbb{\Pi}$:** In these experiments we consider that there is no limitation on the agent's ability to represent value functions, and focus on achieving value equivalence with respect to the polytope of value functions $\ddot{\mathcal{V}}$ induced by the environment. We enable the agent to represent arbitrary functions in $\mathbb{V}$ by restricting ourselves to tabular environments and using dynamic programming to perform exact value iteration in our Policy Construction step. We approximate the value polytope by randomly sampling deterministic policies: $\{\pi_1, \ldots, \pi_n\}$ and evaluating them (again using dynamic programming) to produce $\{v_{\pi_1}, \ldots, v_{\pi_n}\}$. We then choose $\mathcal{V} = \{v_{\pi_1}, \ldots, v_{\pi_n}\}$. In this setting we vary the number of policies generated.

**Corresponding experiments:** the experiments in this class vary two dimensions: (1) the rank of the model and (2) the number of policies generated. In Figures 3(a) and 3(b) we depict plots for the Four Rooms environment that fix the number of policies while varying the rank of the model and plots that fix the rank of the model while varying the number of policies, respectively. Figures 3(c) and 3(d) are analogous plots for the Catch environment.

**span$(\mathcal{V}) \approx \tilde{\mathcal{V}}, \mathbf{\Pi} = \mathbb{\Pi}$:** In these experiments we explore the setting described in Remark 2. We assume that the agent has variable ability to represent value functions, $\tilde{\mathcal{V}}$, and attempt to learn a model in $\mathcal{M}(\tilde{\mathcal{V}}, \mathbb{\Pi})$. From Proposition 1 we only need to find $\mathcal{V}$ such that span$(\mathcal{V}) \supseteq \tilde{\mathcal{V}}$. Experiments in this class can further be broken down into two settings based upon the nature of $\tilde{\mathcal{V}}$:

(a) **Linear function approximation:** In certain experiments our agent uses a class of linear function approximators to represent value functions: $\tilde{\mathcal{V}} = \{\tilde{v} : \tilde{v}(s) = \sum_{i=1}^{d} \phi_i(s) w_i\}$ where $\phi_i(s) : \mathcal{S} \mapsto \mathbb{R}$ and $\boldsymbol{w} \in \mathbb{R}^d$. In this setting achieving span$(\mathcal{V}) \supseteq \tilde{\mathcal{V}}$ can be satisfied by choosing $\mathcal{V} = \{\phi_i\}_{i=1}^{d}$. For experiments using linear function approximation, we select our features $\{\phi_i\}_{i=1}^{d}$ to correspond to state aggregations. This entails the following procedure:

  (i) Collect data using a policy that selects actions uniformly at random.
  (ii) For tabular domains (e.g., Catch, Four Rooms), convert tabular state representations into coordinate-based representations. For Catch we convert each tabular state into the positions of both the paddle and the ball: $(x_{\text{paddle}}, y_{\text{paddle}}, x_{\text{ball}}, y_{\text{ball}})$. For Four Rooms we use the position of the agent: $(x_{\text{agent}}, y_{\text{agent}})$. Denote the function that performs this conversion as: $f : \mathcal{S} \mapsto \mathbb{R}^n$ where $n = 2$ and $n = 4$ for Four Rooms and Catch respectively.
  (iii) Perform k-means clustering on these converted states to produce $d$ centers $c_{1:d}$.
  (iv) Define $\phi_i(s) = \mathbb{1}\{\arg\min_j \|f(s) - c_j\|_2 = i\}$, which corresponds to aggregating states according to their proximity to the previously calculated centers.

  **Corresponding experiments:** the experiments in this class vary two dimensions: (1) the rank of the model and (2) the number of basis functions in $\{\phi_i\}_{i=1}^{d}$. In Figures 4(a) and 4(b) we depict plots of "slices" of this two-dimensional set of results on the Catch domain: 4(a) depicts fixing the number of basis functions while varying model-rank and 4(b) depicts fixing the model-rank while varying the number of basis functions.

(b) **Neural network function approximation:** When Neural Networks are used to approximate the agent's value functions we have $\tilde{\mathcal{V}} = \{\tilde{v} : \tilde{v}(s) = g_\theta(s)\}$ where $g_\theta$ represents a neural network with a particular architecture parameterized by $\theta$. In our experiments we choose the architecture of $g_\theta$ to be a 2 layer neural network with a tanh activation for its hidden layer. Unlike the linear function approximation setting, it is less obvious how to choose $\mathcal{V}$ such that span$(\mathcal{V}) \supseteq \tilde{\mathcal{V}}$. One option is to use randomly initialized neural networks in $\tilde{\mathcal{V}}$ as our basis. To randomly initialize a given layer in some network $g_\theta$, we select weights from a truncated normal distribution where $\mu = 0$ and $\sigma = 1/\sqrt{\text{layer-input-size}}$ and initialize biases to 0.

  However, we found in practice that a large number of these randomly initialized networks were required to achieve reasonable performance. Instead of maintaining a large set of

initializations in $\mathcal{V}$, we allow the elements of $\mathcal{V}$ themselves to be stochastic. Every time we apply an update of gradient descent we sample a new set of randomly initialized neural networks to function as $\mathcal{V}$. This is equivalent to minimizing $\mathbb{E}_\mathcal{V}[\ell_{\Pi,\mathcal{V},\mathcal{D}'}(m^*, \tilde{m})]$ where $\ell_{\Pi,\mathcal{V},D'}$ is defined in 7. We find that having more random elements in $\mathcal{V}$ decreases the variance in the performance of VE models; $|\mathcal{V}| = 5$ in our experiments.

**Corresponding experiments:** the experiments in this class vary two dimensions: (1) the rank of the model and (2) the width of the neural networks in $\tilde{\mathcal{V}}$. In Figures 4(c) and 4(d) we depict plots of "slices" of this two-dimensional set of results on the Catch domain: 4(c) depicts fixing the network width while varying model-rank and 4(d) depicts fixing the model-rank while the network width varies.

### A.2.4   Additional results

In the experimental section of the main text we showed that our theoretical claims about the value equivalence principle hold in practice through a series of bivariate experiments (e.g., varying model-rank and number of bases, varying model-rank and number of policies, varying model-rank and network width). We displayed our results as "slices" of these bivariate experiments, where one variable would be held fixed and the other would be allowed to vary. To keep the paper concise, we only displayed a subset of these slices where the "fixed" variable was selected as the median value over full set we experimented with. In what follows, we present the complete set of the experimental results we acquired. We indicate that a plot was included in the main text by printing its caption in bold font.

(a) Catch (fixed $\mathcal{V}$)   (b) Catch (fixed $\mathcal{V}$)   (c) Catch (fixed $\mathcal{V}$)   (d) **Catch (fixed $\mathcal{V}$)**

(e) Catch (fixed $\mathcal{V}$)   (f) Catch (fixed $\mathcal{V}$)   (g) Catch (fixed $\mathcal{V}$)   (h) Catch (fixed $\mathcal{V}$)

Figure 7: All Catch results with fixed $\mathcal{V}$ and $\mathrm{span}(\mathcal{V}) \approx \tilde{\mathcal{V}}$.

Figure 8: All Catch results with fixed $\tilde{m}$ and $\mathrm{span}(\mathcal{V}) \approx \tilde{\mathcal{V}}$.

Figure 9: All Catch results with fixed $\mathcal{V}$ and $\mathcal{V} = \{v_{\pi_1}, \dots, v_{\pi_n}\}$.

Figure 10: All Catch results with fixed $\tilde{m}$ and $\mathcal{V} = \{v_{\pi_1}, \ldots, v_{\pi_n}\}$.

Figure 11: All Four Rooms results with fixed $\mathcal{V}$ and $\mathcal{V} = \tilde{\mathcal{V}}$.

Figure 12: All Four Rooms results with fixed $\tilde{m}$ and $\mathcal{V} = \tilde{\mathcal{V}}$.

Figure 13: All Four Rooms results with fixed $\mathcal{V}$ and $\mathcal{V} = \{v_{\pi_1}, \ldots, v_{\pi_n}\}$.

(a) Four Rooms (fixed $\tilde{m}$) (b) Four Rooms (fixed $\tilde{m}$) (c) **Four Rooms (fixed $\tilde{m}$)**

(d) Four Rooms (fixed $\tilde{m}$) (e) Four Rooms (fixed $\tilde{m}$) (f) Four Rooms (fixed $\tilde{m}$)

Figure 14: All Four Rooms results with fixed $\tilde{m}$ and $\mathcal{V} = \{v_{\pi_1}, \ldots, v_{\pi_n}\}$.

(a) Cart-pole (fixed $\mathcal{V}$) (b) Cart-pole (fixed $\mathcal{V}$) (c) **Cart-pole (fixed $\mathcal{V}$)**

(d) Cart-pole (fixed $\mathcal{V}$) (e) Cart-pole (fixed $\mathcal{V}$) (f) Cart-pole (fixed $\mathcal{V}$)

Figure 15: All Cart-pole results results with fixed $\mathcal{V}$ and $\mathrm{span}(\mathcal{V}) \approx \tilde{\mathcal{V}}$.

### A.2.5 Hyperparameters

Table 1 provides a list detailing the different hyperparameters used throughout our pipeline.

Figure 16: All Cart-pole results results with fixed $\tilde{m}$ and $\mathrm{span}(\mathcal{V}) \approx \tilde{\mathcal{V}}$.

| Hyperparameter | Value | Description |
|---|---|---|
| minibatch size | 32 | Number of samples passed at a time during a training step of any learning method. |
| model learning rate | 5e-5 | Learning rate used to train all models. |
| # model samples | 1,000,000 | Number of transitions sampled by a random policy in the Data Collection phase. |
| model depth | 2 | Number of hidden layers in the model architecture. |
| model width | 256 | Number of units per hidden layer. |
| model activation | tanh | Activation function following a hidden layer. |
| model learning max steps | 1,000,000 | Maximum number of training iterations. |
| $\gamma$ | 0.99 | Discount factor used across environments. |
| LSTD samples / policy | 10,000 | Number of samples collected for each phase of policy evaluation using LSTD. |
| # policy iteration steps | 40 | Number of steps of policy iteration in the policy construction phase, when applicable. |
| DQN learning rate | 5e-4 | Learning rate for DQN. |
| DQN # environment steps | 2,500,000 | Number of environment steps that DQN learns over. |
| DQN learning frequency | 4 | A learning update is applied after this many environment steps. |
| DQN depth | 1 | Number of hidden layers in the DQN. |
| DQN activation | tanh | Activation function following a hidden layer. |
| DQN target update | 2000 | Number of environment steps before the target network in the DQN is updated. |
| Tabular # eval episodes | 20 | Number of episodes to average performance over to assess a policy in the tabular setting. |
| DQN # eval episodes | 100 | Number of episodes to average DQN policy performance over at the end of training. |
| DQN $\epsilon$ | 0.05 | Chance of picking a random action during training. |
| Optimizer | Adam | Optimizer used for all learning operations. Default Adam parameters were used. |

Table 1: List of hyperparameters used in the experiments.