[Reviews · NeurIPS 2020]

Review 1

Summary and Contributions: Thank you for the rebuttal. The authors have responded to one of my main concerns, the lack of related work, by stating that the references provided are not relevant. Many of the deep learning papers included learn not only the state representation but also the model. Therefore, treating these references as not relevant as the authors do is incorrect. Additionally, given that the main contribution is theoretical, by not discussing how more empirical work fits this framework, the paper is incomplete. One thing I am more convinced by after rebuttal is that in this case a tabular setup might be enough for an empirical evaluation. As such, I will increase my score, but would still recommend rejection. This paper proposes the use of value equivalence as a principle for learning representations in Reinforcement Learning problems. It includes some work on the properties of this method, and links the idea back to works that use differentiable planning networks. It includes experimental evaluation on mostly tabular methods, comparing to an internal baseline.

Strengths: - The paper provides a theoretical analysis of the proposed value equivalence and how the model class collapses to the true model as more policies are taken into consideration. - The paper provides enough details in the appendix to be reasonably reproducible

Weaknesses: - The paper does not do a good job of placing itself with regards to earlier work in this field, missing most of the field of state abstraction in RL, i.e. Chapman 1991, Dean 1997, Givan 2002, Ravindran 2003, Ferns 2004, Li 2006, Taylor 2009 - It has also missed most recent Deep Learning approaches to state/model learning in RL i.e. Watter 2015, Igl 2018, Corneil 2018, Francois-Lavet 2019, M. Zhang 2019, A. Zhang 2019, Gelada 2019, Castro 2019, Biza 2019, van der Pol 2020. - There is a very short paragraph on a very limited subset of these methods in the Appendix (which is easy to miss, as there is no reference to this literature section in the paper!), and even there this work is not discussed seriously. This paper is not complete without a serious discussion of these methods, including broader experimental comparisons. - The proposed equivalence relation seems very similar to Q-pi-equivalence as discussed in Li, 2006, with some additional theoretical exposition. I would like to hear from the authors what the difference is between their notion of value-equivalence and the one mentioned in Li, 2006. - The paper contains references to “usually” or “the common practice” without any citations to other research papers to back up the claim. I would like the authors to corroborate these claims with citations to relevant works. - Additionally, the experimental section is limited and not clearly presented. In particular, tabular methods are used for two out of three domains, but this is only mentioned in the Appendix. This is surprising, because learning representations is especially interesting for problems where the state space cannot reasonably be represented by a tabular method. Since the authors do not motivate this choice in the paper, I would like to hear their motivation for choosing tabular methods. References: Chapman, David, and Leslie Pack Kaelbling. "Input Generalization in Delayed Reinforcement Learning: An Algorithm and Performance Comparisons." 1991. Dean, Thomas, and Robert Givan. "Model minimization in Markov decision processes." 1997. Givan, Robert, Thomas Dean, and Matthew Greig. "Equivalence notions and model minimization in Markov decision processes." (2003) Ravindran, Balaraman, and Andrew G. Barto. "Approximate homomorphisms: A framework for non-exact minimization in Markov decision processes." (2004). Ferns, Norm, Prakash Panangaden, and Doina Precup. "Metrics for Finite Markov Decision Processes." 2004. Li, Lihong, Thomas J. Walsh, and Michael L. Littman. "Towards a Unified Theory of State Abstraction for MDPs." 2006. Taylor, Jonathan, Doina Precup, and Prakash Panagaden. "Bounding performance loss in approximate MDP homomorphisms." 2009. Watter, Manuel, et al. "Embed to control: A locally linear latent dynamics model for control from raw images." 2015. Igl, Maximilian, et al. "Deep variational reinforcement learning for POMDPs." 2018 Corneil, Dane, Wulfram Gerstner, and Johanni Brea. "Efficient model-based deep reinforcement learning with variational state tabulation." 2018 François-Lavet, Vincent, et al. "Combined reinforcement learning via abstract representations." 2019. Zhang, Marvin, et al. "SOLAR: Deep structured representations for model-based reinforcement learning." 2019. Zhang, Amy, et al. "Learning causal state representations of partially observable environments." 2019 Gelada, Carles, et al. "Deepmdp: Learning continuous latent space models for representation learning." 2019 Castro, Pablo Samuel. "Scalable methods for computing state similarity in deterministic Markov Decision Processes." 2019 van der Pol, Elise, et al. "Plannable Approximations to MDP Homomorphisms: Equivariance under Actions." 2020 Biza, Ondrej, et al. "Learning discrete state abstractions with deep variational inference." 2020

Correctness: The experimental section is not convincing, and a more reasonable comparison between your work and one or two methods used in the field seems necessary. - For most experiments, the authors use tabular models, which is not mentioned in the paper’s experiment section but only discussed in the Appendix. Things are often much simpler in the tabular case (e.g. contrast tabular Q-learning with DQN) but representation learning is often required for problems where tabular approaches are not feasible. - For NNs, which are only used for Cartpole, the “MLE” approach entails the MSE between a state and the next state. For VE, the authors use the value difference. This is not a fair comparison, since the baseline is likely to suffer from the ‘trivial embedding’ problem (discussed in Francois-Lavet 2019, Gelada 2019, van der Pol 2020), which is resolved in a multitude of ways in the related work. Comparing to an approach from the literature, would make the comparisons more convincing. Questions about the theory: - Line 131: “adding more values to V”, V is supposed to be a set of functions according to Definition 1, now it is a set of values? - Line 138: now V is a set of functions. Please clarify - Line 197: the authors claim that l_p is usually defined on the principle of MLE. This should be supported with references. - Line 201: Please state which are those desirable properties and why they are not necessary anymore. - Line 274: Can you clarify when linear function approximators are ‘as good as any model’? - Equation 7: It is not clear how to use the loss function in Eq. 7, since one cannot reasonably sum over all possible value functions and policies. In the experiments the authors do not do that (line 286) - The space of policies is continuous, but the paper is now only considering deterministic policies that always take the same action in every state. Please clarify. -What is the difference between the proposed value-equivalence and that described in Li, 2006?

Clarity: The paper is reasonably readable, although some things are unclear. Some comments (in addition to the above questions) below: - Line 12/Line 42: What does “augment” mean in this context? - Line 29: “A common practice in model-based RL is […]”, this claim should be supported with references to the literature. - Property 3: Is this meant to reflect ‘adding more policies reduces the set of models that explains them’? Please clarify. - Terms such as 'Hamel-dimension' should be introduced before use.

Relation to Prior Work: While the relation to different types of value-networks in the Table of line 338 is nice, there is no mention at all to a huge body of related work (see Section 'Weaknesses') in the main paper, and only a cursory mention in the Appendix.

Reproducibility: Yes

Additional Feedback:


Review 2

Summary and Contributions: This paper's central thesis is that in model-based reinforcement learning, the main thing we need from a learned model is often accurate value estimates, and accurate state predictions are merely a proxy for that ultimate goal. As such, it establishes a theoretical concept called "value equivalence" which applies to two models that yield the same Bellman backups for a given set of policies and value functions. The main practical upshot of this is that if the space of policies and value functions that the model will operate on is limited, then it may be possible to learn a simpler model that is value equivalent to the real environment over that limited policy/value space. The paper proves a number of facts about value equivalence and proposes a loss function for training a model that encourages value equivalence. Small-scale experiments show that models learned in this way can yield better planning performance. The paper also suggests that the principle of value equivalence may be at play in several recent model-based approaches.

Strengths: - The theoretical framework presented here is, as far as I know, novel and seems interesting and insightful. It is intuitive that an agent that can't do very much probably doesn't need a very complicated model -- I appreciate that this paper takes that intuition and formalizes it and then uses it to derive a novel loss function for model learning that accounts for this principle. - The theoretical exploration of this concept establishes a solid foundation for reasoning about when it is likely to offer useful insight and/or practical advantage. - The experiments illustrate that it is possible to apply the derived loss function in practice to improve MBRL performance. - The work has the potential to improve scientific understanding of the strengths and weaknesses of MBRL algorithms as well as inspire novel MBRL algorithms that can learn a model specific to the class of policies/value functions the agent might learn.

Weaknesses: - The empirical aspect of this paper is fairly thin, more of a proof of concept. Given the strength of the theoretical/conceptual component, I think that's okay, but it's certainly true that the paper would be stronger with a more robust empirical evaluation that more compellingly demonstrated the practicality of this approach. - I found Section 6 to be difficult to follow because it is so compressed. Though the technical content is there, one would have to be deeply familiar with these algorithms to be able to evaluate these claims or gain meaningful insight from this section. I understand that space is limited and would definitely prioritize clear development of the core ideas over this discussion. However, the authors might consider alternative models. Maybe it would be more valuable to describe the connection to one of these existing algorithms with the same clarity and patience as the sections that precede it, and leave the details of the others to an appendix? - While the paper does discuss how it improves understanding of recent existing work, I think it misses some related prior work that touches upon these ideas (see the Relation to Prior Work section).

Correctness: I am not aware of any technical errors in this paper.

Clarity: I found this paper to be exceptionally clear and well-written. I really appreciated the clear presentation of the theoretical results, which successfully mixes formal and intuitive development of the ideas.

Relation to Prior Work: The paper does a good job of surveying relevant recent work, especially in Deep MBRL where several approaches have attempted to learn a task-specific model. However, I think the paper should discuss existing work that is conceptually similar in formalizing the relationships between the representations of models, policies, and value functions. Here are a couple of examples that come to mind: - I'm surprised that the paper does not seem to discuss model minimization and specifically MDP homomorphism (Ravindran and Barto 2002 and then a whole lot of follow-up work), which similarly aims to induce a model simpler than the true dynamics that nevertheless produces the same value function/policy. It's clear to me that value equivalence is a distinct concept, but it seems important to discuss the distinctions. - Another existing result that seems highly related is the equivalence of the best linear approximate value function in a perfect model and the optimal policy in the best linear approximate model (concurrently reported on by Sutton et al. 2008 and Parr et al. 2008). Here is an illustration of the connection between the model class and value function class, where a limitation in either allows one to narrow the space of the other. It may be that this result could be re-derived using the more general concept of value equivalence.

Reproducibility: Yes

Additional Feedback: After author response: Thank you to the authors for the response. My comments were adequately addressed. I will re-emphasize how important it is for the paper to clearly address state abstraction. I do understand that the framework in this paper is distinct and interesting in its own right, but it is important for the paper to clearly discuss the similarities and differences for the sake of a reader who wants to know where this work fits and what it adds. Some of the other reviewers would like to see a more robust survey of model-based methods in general. Personally, I think that this paper is addressing a particular theoretical question, so it doesn't need to discuss every paper that just uses a model. However, it is important that the authors consider those references and incorporate discussions of related ideas contained therein.


Review 3

Summary and Contributions: 1. The paper provides an interesting mathematical characterization of decision-aware model learning for value-based RL. The overall high-level idea of value equivalence is straightforward, yet as far as I know, the concept of "value equivalence" as formulated in terms of a general set of policies and value functions is indeed novel and complementary to the value-aware model learning (VAML) theoretical analysis of Farahmand. 2. The contributions are mainly theoretical (e.g., properties of the space of value-equivalent models and a taxonomy that reinterprets other model-based algorithms -- VINs, Predictron, TreeQN, MuZero -- under the concept of value equivalence). 3. Though the theoretical formulation does not directly translate into novel algorithmic insights, the proposed understanding of a basis for the space of value-equivalent models may help identifying good model classes in the future conditioned on which planning algorithm will use the models. 4. Though all proofs are relegated to the appendix, I found that the intuitions provided can help the reader to follow (not all but) most derivations. 5. A number of small-scale experiments confirm previous results on improved performance over MLE-based models (in particular, when dealing with a restricted model class).

Strengths: See summary.

Weaknesses: See detailed comments below.

Correctness: No concerns.

Clarity: With a few exceptions where clarification is needed (see detailed comments), yes.

Relation to Prior Work: Important discussion of previous work on VAML is omitted from the main paper (see detailed comments).

Reproducibility: Yes

Additional Feedback: POST-REBUTTAL Thanks for the well-argued rebuttal; my overall recommendation is to accept though I will keep my score at 6 because I believe the present submission can do a much better job of discussing related work. I believe the authors were incorrect to refer to R1's suggestions as a misunderstanding; in fact, their own response suggests just the opposite: "It seems plausible that the model learned by VE induces a space of 'compatible' state representations". This is *exactly* what I thought after considering the relation of these suggested citations (that I wish I had thought to mention myself)... I think there might be a very simple "reduction" between these two approaches (at least in one direction) that extends the theoretical results of the paper well beyond the present scope. Everyone wins in this case and I strongly encourage the authors to consider these issues more carefully on revision. In reading the other reviews and in light of my own, I think it is important simply to reiterate to the authors the importance of the following two key revisions: (1) Moving some key related work to the main paper as noted in the reviews. (2) For completeness, making an extended related work discussion including all citations from reviewers in the Appendix (if not in the main paper). While some references are more distantly related than others, I believe all are relevant for discussion, even if some are broadly grouped into general classes of approaches. ============================================================== PRE-REBUTTAL Overall, I found the paper to have a solid theoretical grounding and in general I think this is an important research direction. However, I would like to see a few points addressed before this paper is published: 1. The scholarship must be improved. The appendix provides a good literature review, but the main text only briefly touches upon important related work. In particular, I strongly feel that the authors should make clear in the paper the differences of their proposed value-equivalence formulation w.r.t. VAML. As a matter of fact, they have already done that in the appendix (i.e., "[...] while we focus on the characterization of the space of value-equivalent models, VAML focuses on the solution and analysis of the induced optimization problem." (lines 839 - 841)), but for some reason they decided not to keep this discussion in the main text. Furthermore, the current discussion of VAML is too minimal -- it should be briefly reviewed so as to better understand the technical differences w.r.t. the contributions of this work. 2. As I said earlier, the theoretical development is not hard to follow with the help of the examples and intuitions provided. The only exception for me was the definition of \mathcal{P}(\Pi, \mathcal{V}) used in Proposition 3 (line 205) which was not entirely clear to me from the following explanations provided in lines 206 - 210 and Figure 2. In particular, I don't get the dependence on the set of value functions \mathcal{V}. 3. Also, related to previous comment (2), I found it hard to follow the developments in lines 167 - 175 leading to Proposition 2. Maybe this is due to my limited background in statistical learning, but relying on the Hamel dimension as if it were some trivial concept broke my reading flow; even the formal derivations in the appendix didn't quite help. Maybe this is something well-known for theorists, but given that the paper strives to make everything easy to understand for the general audience with examples and intuitions, I would appreciate it if the authors could explain/motivate this Hamel dimension a little bit better. All in all, I liked the paper. While honestly, I am not overwhelmed with what I learned, I still believe it is an important and clean elaboration on the topic of decision-aware model learning and I appreciate the fact that multiple authors are currently putting this topic on a firm theoretical foundation. Nonetheless, because VAML has previously investigated this topic, it is absolutely critical to prominently discuss that work in the main paper and how the contributions of this paper differ from that paper.


Review 4

Summary and Contributions: Post Rebuttal: As mentioned by R1, the state representation literature is actually quite relevant here. One key advantage of recovering the state representation is to use it to learn a model on top of it. An example of this work is: Provably efficient RL with Rich Observations via Latent State Decoding, Du et al. ICML 2019. This paper will benefit from properly situating itself in this literature. I understand that this is a daunting task since the MBRL and state representation literature is huge, but a detailed comparison with the most relevant and an acknowledgement of the rest should suffice. ----------- Paper proposes a value-equivalence approach for model-based reinforcement learning (MBRL). The key idea is that for a given set of policy and value functions; if two models (reward and dynamics) give the same Bellman updates then we can use them interchangeably (such models are called value-equivalent). This is helpful, since it maybe possible to learn a simpler model which is value-equivalent to the gold model. Most of the paper proves property of value-equivalence and proposes heuristics to applying it in practice. Experiments are presented on a set of domains showing its promise.

Strengths: I think the intuition with value-equivalence is spot on. Maximum likelihood estimation ignores value and policy information, and tries to accomplish more than what is needed. This insight has been argued in previous work (e.g., Farahmand et al., 2017 and 2019). This paper continues this thinking and develops a mature theory. I found the paper to be well-written and connection to prior work in Section 6 is interesting.

Weaknesses: I am worried about the practicality of this system. I much like the value-aware model loss in Farahmand et al., 2017; or you learn model weighted by V(s') so you allow the model to be more noisy where the value of your current policy is lower. This does not require defining any set of policy or model. The paper discusses a set of approaches in Section 4.1 but these are for special cases. Authors do discuss a sampling heuristic in line 279-281.

Correctness: I tried to verify several results and I believe they are sound. I have a question: - How can m* not be inside M(\Pi, \mathcal{V})? As defined on line 97, as the value-equivalence is an equivalence relation and M(\Pi, \mathcal{V}) is the set of models which are value-equivalent to m*, therefore, M(\Pi, \mathcal{V}) should always contain m*. Therefore, I am not sure why property 2 or remark 1 talk about the set being empty.

Clarity: The paper is well written.

Relation to Prior Work: Paper mostly do justice to prior work (at least the papers most related to this approach are cited). However, citations are missing in DeepRL and RL theory area. E.g., Model-Based Reinforcement Learning for Atari (Kaiser et al.) in Deep RL, and Model-based RL in Contextual Decision Processes: PAC bounds and Exponential Improvements over Model-free Approaches (Sun et al. 2018) in RL theory.

Reproducibility: Yes

Additional Feedback:

[Author Response · NeurIPS 2020]

Thank you very much for the thoughtful reviews!

2
**All reviewers**: All reviewers suggested ways to improve our treatment of the related work. If the paper is accepted we
will use the additional content page to address this. We will move (a revised version of) Sec. A.3 from the appendix
to the main paper and make sure the connections with the literature are explored in more depth (in particular with
respect to VAML, as suggested by **R3** and **R4**, and several notions of MDP homomorphism, as suggested by **R1** and **R2**).

7
[**R1**]: We want to emphasize that our work is about model learning, *not* about state representation. In particular, we
assume (except in the discussion of related work) the existence of a state signal $s$ that is neither learned nor modified
by VE. The VE loss is used exclusively to learn a model from state $s$ to state $s'$, but the notion of state itself never
changes. It seems plausible that the model learned by VE induces a space of "compatible" state representations, but this
is yet to be analyzed. We made a deliberate choice to isolate the representation learning aspect, both in the theory and
experiments, to have as clean an analysis as possible. We believe that this misunderstanding may be the source of **R1**'s
concerns. In light of this clarification, we kindly ask the reviewer to reconsider their assessment of our paper.

We argue that a tabular representation is the most appropriate setup to understand a new approach to model-learning. In
particular, tabular experiments help to illustrate the differences between VE and MLE, since in this case it is trivial
to define a distribution model with the appropriate capacity (a $n \times n$ transition matrix) and to limit its capacity in a
meaningful way (through the matrix's rank). We also scale up to non-tabular experiments (Figs. 4c and 4d) and consider
the relationship to existing work that uses VE in combination with deep learning (Sec. 6). The choice of MLE as our
baseline also arises naturally, as this is by far the criterion most commonly used in conventional model learning (see
supplement of [1]). Also, the "trivial-embedding" problem cannot be the explanation for the poor performance of MLE,
since there is no embedding being learned (states $s$ are fixed; only the transition function $f(s, a) = s'$ is learned). We
believe other concerns raised by **R1** can be similarly resolved by noting that VE is not about representation learning.

**Specific points:** ($Q^\pi$**-irrelevance**): see response to **R2** on homomorphisms. (**L131/138**): We will use "functions"
throughout, thanks for pointing that out! (**L197 & L29**): We will include references to standard RL texts that corroborate
these claims. (**L201**): MLE generates probability distributions under which the observed data is most probable [2]; the
fact that this is desirable does not mean MLE will always be the best strategy. (**L274**): The statement does not refer to the
linear approximators themselves, but rather to models (linear or otherwise) that are VE with respect to them (see L277).
(**Eq 7**): The loss in Eq. 7 is over (possibly very small) subsets of all possible functions and policies; this is exactly how
VE is used to restrict the space of models and one of the main points of the paper. (**Policies**): We consider stochastic
policies throughout, but describe after Proposition 1 how $|\mathcal{A}|$ deterministic policies can cover this space. (**L12/42**):
It refers to adding functions / policies to the sets we are defining VE wrt. (**Property 3**): Adding more policies and
functions to $\Pi$ and $\mathcal{V}$ further constrains the set of models that are VE to the true model. (**Hamel**): see comment to **R3**.

[**R2**]: If the paper is accepted, we will use the extra space to improve the discussion on related work. We now describe 3
concrete modifications in this direction resulting from **R2**'s comments. (**Sec. 6**): We will revise Sec. 6 prioritising clarity
over breath, to make sure the main text is self-contained. (**Homomorphism**): We will elaborate on the connection
between VE and MDP homomorphisms, which we briefly touched upon in Sec. A.3. Note that any notion of equivalence
over states (*e.g.*, $Q^\pi$-irrelevance, suggested by **R1**) can be recast as a form of state aggregation; in this case the functions
mapping states to clusters can (and probably should) be used to enforce VE. But VE is more general than that: it applies
to any representation (not only aggregation) and can be used to explore structure in the problem even when there is
no clear notion of state abstraction (Sec. A.1.2). We will add this discussion. (**Linear models**): The relation of VE
with Sutton *et al.*'s and Parr *et al.*'s results is another interesting connection. As **R2** notes, VE is more general, since it
also applies to nonlinear models (even when a linear approximator is used—L277). Re-deriving these results from
VE's perspective is an intriguing idea; we will try to do so and add any eventual insights to the appendix.

[**R3**]: We will extend the discussion on VAML and move it to the main paper. (**Clarification of** $\mathcal{P}(\Pi, \mathcal{V})$): We use $\mathcal{P}$
to refer to the set of all transition kernels, and define $\mathcal{P}(\Pi, \mathcal{V})$ as the set of all such kernels that are value equivalent
to the environment wrt $\Pi$ an $\mathcal{V}$ when the reward is assumed to be known. Upon re-examination, it appears that we
do not explicitly state this until the proof of Proposition 2 in the appendix. We will spell out this definition clearly
in the main text in the subsequent version of the paper. (**Hamel**): Hamel dimension, matching the intuitive notion
of dimensionality, describes the number of coordinates necessary to specify every point in a given vector-space. We
will include a definition of this term in the text to improve readability.

[**R4**]: (**Practicality**): This is a valid point. However, note that we empirically observed that high quality value equivalent
models can be produced without requiring prohibitively many value functions. We intend to provide theoretical support
for these observations in future work. (**Property 2**): Good question! $\mathcal{M}(\Pi, \mathcal{V})$ is the set of models in class $\mathcal{M}$ that
are value-equivalent to $m^*$. When we consider all policies and values, there is only one possible value equivalent
model: $m^*$ itself. Thus, if $m^* \notin \mathcal{M}$ then $m^* \notin \mathcal{M}(\mathbb{\Pi}, \mathbb{V})$. (**Related Work**): Thanks for pointing these out! We will
include both of these papers in the next version of our related work section.

[1] Farahmand et al., 2017. Value-Aware Loss Function for Model-based Reinforcement Learning. [2] Millar, 2011. Maximum Likelihood Estimation and Inference.


[Meta-Review · NeurIPS 2020]

The paper introduces the concept of Value Equivalence (VE) in the model-based RL. VE states that two models are equivalent w.r.t. a set of value functions F and policies Pi if the effect of the Bellman operator for policies in Pi on the value functions in F are the same. The paper theoretically studies several aspects of the VE concept. This is a good theoretical work in MBRL, and especially in what might be called the problem of decision-aware model learning. Three out of four reviewers are in favour of accepting this paper. There are, however, certain concerns, some of which I briefly summarize below. Most of the concerns are related to how this paper positions itself in relation to the current literature. These issues are important, and the current version of the paper doesn’t do a very good job in acknowledging and positioning itself. We had a lot of discussions among reviewers, as well as between AC and SAC. I believe that with an honest effort, the authors can revise the paper and address these concerns without requiring another round of review. So I recommend acceptance of this work trusting the authors to seriously consider these in revising their paper. - The goal of the paper is very similar to the goal of the Value-Aware Model Learning (VAML) framework (as well as more recent Gradient-Aware Model-based Policy Search and Policy-Aware Model Learning methods), though their focus are somewhat different. It is important to emphasize and discuss this relation to VAML more prominently in the main body of the paper, instead of deferring it to an appendix. - The literature on the state abstraction is relevant to this work, even though their goals are not exactly the same. This is especially the case for deep RL approaches that learn both a state abstraction and a model. - It is suggested that the experimental part of the paper could be improved. This isn't critical, but it would be a welcome improvement.